# Prior probability cues bias sensory encoding with increasing task exposure

Kevin Walsh[1]*, David P McGovern[2], Jessica Dully[3], Simon P Kelly[4,5], Redmond G O'Connell[5,6]

[1]School of Psychological Sciences, Monash University, Melbourne, Australia; [2]School of Psychology, Dublin City University, Dublin, Ireland; [3]Institute of Psychiatry, Psychology & Neuroscience, King's College London, London, United Kingdom; [4]School of Electrical Engineering, University College Dublin, Dublin, Ireland; [5]Trinity College Institute of Neuroscience, Trinity College Dublin, Dublin, Ireland; [6]School of Psychology, Trinity College Dublin, Dublin, Ireland

*For correspondence:
kevin.walsh1@monash.edu

**Abstract** When observers have prior knowledge about the likely outcome of their perceptual decisions, they exhibit robust behavioural biases in reaction time and choice accuracy. Computational modelling typically attributes these effects to strategic adjustments in the criterion amount of evidence required to commit to a choice alternative - usually implemented by a starting point shift - but recent work suggests that expectations may also fundamentally bias the encoding of the sensory evidence itself. Here, we recorded neural activity with EEG while participants performed a contrast discrimination task with valid, invalid, or neutral probabilistic cues across multiple testing sessions. We measured sensory evidence encoding via contrast-dependent steady-state visual-evoked potentials (SSVEP), while a read-out of criterion adjustments was provided by effector-selective mu-beta band activity over motor cortex. In keeping with prior modelling and neural recording studies, cues evoked substantial biases in motor preparation consistent with criterion adjustments, but we additionally found that the cues produced a significant modulation of the SSVEP during evidence presentation. While motor preparation adjustments were observed in the earliest trials, the sensory-level effects only emerged with extended task exposure. Our results suggest that, in addition to strategic adjustments to the decision process, probabilistic information can also induce subtle biases in the encoding of the evidence itself.

## eLife assessment

This **important** paper sheds light on the role of expectations in perceptual decision-making. Sophisticated analyses of human EEG data provide **convincing** evidence that both motor preparation and sensory processing were affected by expectations, albeit with different time courses. These findings will be of interest to scientists interested in perception and decision-making.

## Introduction

During perceptual decisions, competing perceptual hypotheses are evaluated based on a combination of sensory data and prior knowledge. When observers are provided with predictive information about the correct choice, they exhibit behavioural biases favouring the more probable alternative, characterised by faster and more accurate decisions when the stimulus matches expectations (*Summerfield and de Lange, 2014*). How these expectations are encoded and integrated into the decision process is a critical question in cognitive science and a subject of ongoing debate (*Deneve, 2012*; *Friston,*

*2010*; *Gold and Stocker, 2017*; *Heeger, 2017*; *Pouget et al., 2013*; *Press et al., 2020*; *Summerfield and de Lange, 2014*; *Teufel and Fletcher, 2020*).

A key question dominating recent work in this field has been whether the influence of prior probability on our choice behaviour reflects purely strategic adjustments or also incorporates biases in the encoding or weighting of the sensory evidence that informs our decisions. Traditional accounts of the computational architecture of visual cortex emphasise a feedforward processing flow (*Crick and Koch, 1998*; *Koch and Poggio, 1999*) and tend to assume that early sensory processing exhibits a high degree of fidelity to the physical stimulus with little or no 'cognitive penetration' (*Pylyshyn, 1999*). Indeed, optimality research has demonstrated that prior knowledge can be incorporated into the decision process through strategic adjustments to decision criteria alone (*Bogacz et al., 2006*; *van Ravenzwaaij et al., 2012*) and mathematical models implementing such adjustments have been shown to comprehensively account for prior-informed behaviour (e.g. *Leite and Ratcliff, 2011*; *Mulder et al., 2012*; *Mulder et al., 2014*; *Ratcliff and McKoon, 2008*; *Ratcliff and Smith, 2004*). Neurophysiological research has provided further support for the role of criterion adjustments with numerous studies demonstrating elevated starting levels of motor preparation for the predicted response alternative (e.g. *de Lange et al., 2013*; *Kelly et al., 2021*). However, several modelling studies have indicated that these adjustments are accompanied by additional biases in the rate of evidence accumulation for the expected alternative at the decision-level (e.g. *Dunovan et al., 2014*; *Hanks et al., 2011*; *Kelly et al., 2021*; *van Ravenzwaaij et al., 2012*; *Wyart et al., 2012*). This raises the question of whether expectation-based effects in perceptual decision-making may in part arise from biases in the original encoding of the sensory evidence feeding the decision process.

An extensive recent literature spanning multiple species and neural assays has examined prior probability effects on sensory processing (*de Lange et al., 2018*; *Feuerriegel et al., 2021b*; *Heilbron and Chait, 2018*; *Walsh et al., 2020*) and a substantial number have provided compelling evidence of a variety of anticipatory and stimulus-evoked sensory-level modulations in humans (e.g. *Aitken et al., 2020*; *Egner et al., 2010*; *Ekman et al., 2017*; *Esterman and Yantis, 2010*; *Grotheer and Kovács, 2015*; *Kok et al., 2012*; *Kok et al., 2013*; *Kok et al., 2014*; *Puri et al., 2009*; *Richter et al., 2018*; *Trapp et al., 2016*). However, many of the key studies demonstrating expectation-based modulations of sensory activity in humans rely on voxel-based analyses of BOLD signals, where there is a limited understanding of the relationship between the macroscopic signal dynamics and the underlying processing of sensory evidence (*Logothetis, 2008*). There is evidence of expectation effects on sensory processing in electrophysiological research on non-human primates (e.g. *Kaposvari et al., 2018*; *Meyer and Olson, 2011*; *Meyer et al., 2014*; *Ramachandran et al., 2017*; *Schlack and Albright, 2007*; *Schwiedrzik and Freiwald, 2017*) and rodents (e.g. *Findling et al., 2023*; *Fiser et al., 2016*; *Gavornik and Bear, 2014*), but the evidence from human electrophysiology is more mixed (e.g. *Aitken et al., 2020*; *den Ouden et al., 2023*; *Feuerriegel et al., 2018*; *Hall et al., 2018*; *Kok et al., 2017*; *Rungratsameetaweemana et al., 2018*; *Solomon et al., 2021*; *Stefanics et al., 2014*; *Tang et al., 2018*). While some have reported expectation effects in humans using EEG/MEG, these studies either measured sensory signals whose relevance to the decision process is uncertain (e.g. *Blom et al., 2020*; *Solomon et al., 2021*; *Tang et al., 2018*) and/or used cues that were implicit or predicted a forthcoming stimulus but not the correct choice alternative (e.g. *Aitken et al., 2020*; *Feuerriegel et al., 2021c*; *Kok et al., 2017*). To assess whether prior probabilities modulate sensory-level signals directly related to participants' perceptual decisions, we implemented a contrast discrimination task in which the cues explicitly predicted the correct choice and where sensory signals that selectively trace the evidence feeding the decision process could be measured during the process of deliberation.

The task was to discriminate the relative contrast of two overlaid gratings, so both gratings were equally task-relevant and the cues provided no spatial information. In addition, the task was designed to be difficult and to require extended deliberation, thus facilitating examination of sensory encoding responses before and during decision formation. We traced sensory encoding dynamics via the contrast-dependent steady-state visual-evoked potential (SSVEP; reviewed by *Norcia et al., 2015*), which is thought to originate in early visual cortex (*Di Russo et al., 2007*; *Lauritzen et al., 2010*; *Vanegas et al., 2013*) and is necessarily driven by units tuned to the stimulus, as only these units would be expected to respond at the specific frequency of the eliciting stimulus modulation. By flickering the two overlaid contrast gratings at different frequencies, it was possible to isolate a separate

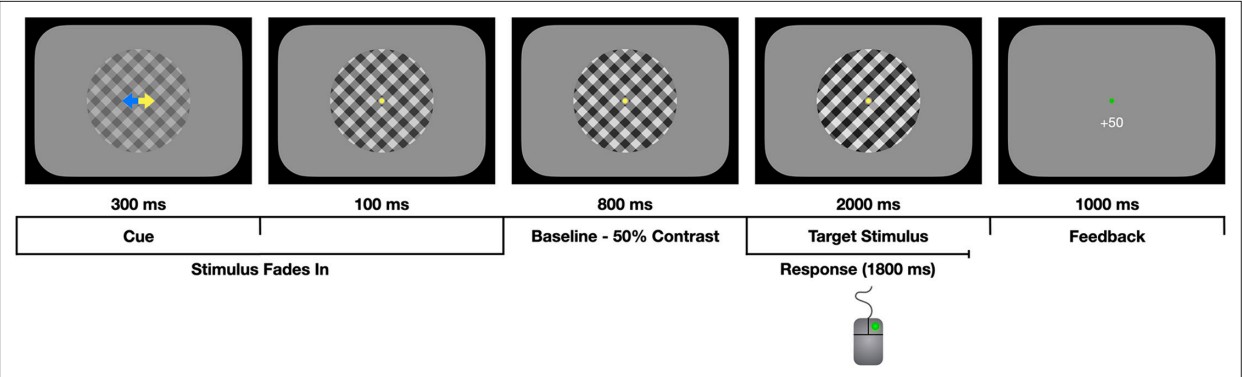

**Figure 1.** Task schematic. Participants were asked to decide which of two overlaid gratings (left-tilt or right-tilt) were presented at a higher contrast while fixating on a central fixation dot. The gratings gradually faded in from 0% to 50% contrast during the first 400 ms, as the predictive cue was presented. The cue (yellow arrow) could be valid, invalid, or neutral (both arrows yellow). The gratings were then held at 50% contrast for a 800 ms baseline phase. At evidence onset, there was an instantaneous increase in the contrast of the 'target' grating (right-tilt in the illustration above) and a reciprocal decrease in the contrast of the 'non-target' grating. The participant responded by clicking the left or right mouse button and they received feedback on their choices in the form of points and the fixation dot changing colour to green (correct) or red (incorrect/early response/miss).

SSVEP for each grating and to measure their differential amplitude which would reflect the sensory quantity informing the perceptual choice. In addition to its excellent contrast sensitivity, the amplitude of SSVEP signals have been shown to predict choice behaviour on tasks where participants are asked to detect or discriminate stimulus contrast changes (e.g. *Grogan et al., 2023*; *O'Connell et al., 2012*; *Steinemann et al., 2018*), validating its characterisation as an index of evidence encoding.

We sought to determine if predictive cues lead to changes in sensory processing that could potentially contribute to the behavioural biases that expectations induce in perceptual decision-making. Given the more consistent observation of sensory-level prior probability modulation in studies of non-human animals that require extensive training, an additional goal of this study was to determine how any such sensory modulations might evolve as a function of extended task exposure. Here, we investigated if the magnitude of any modulation of sensory encoding associated with prior probabilities was enhanced as the participant became increasingly adept at contrast discrimination with practice and whether an emerging bias in sensory encoding may result in subtle adjustments to the incorporation of probabilistic information in the decision strategy.

## Results

We recorded 128-channel EEG data from 12 human participants performing a two-alternative forced-choice contrast discrimination task, where they decided which of two orthogonally oriented (±45° from vertical) overlaid gratings was presented with a higher contrast on discrete trials (see *Figure 1*). Across three to five sessions, participants completed 5750 trials on average. Each trial contained one of three predictive cues before evidence onset: valid cues correctly identified the tilt of the target stimulus on the subsequent trial; invalid cues indicated the opposite of the target tilt; and neutral cues provided no information about the likely answer. Contrast was individually titrated to achieve 70% accuracy and points were awarded/deducted for correct (+50 points) or error (-25 points) responses. Participants were instructed to maximise their score and respond as soon as they were quite sure of their decision by clicking the mouse button (left or right) corresponding to the tilt direction of the chosen grating. We evoked SSVEPs by separately 'frequency-tagging' the gratings by reversing their spatial phase at 20 and 25 Hz. This assignment of these frequencies to each grating was randomly counterbalanced across trials.

### The effect of the cue on behaviour

Participants exhibited the behavioural biases in reaction time and accuracy that are typically observed in response to prior probability cues. An initial full mixed effects model analysis of reaction time, including choice accuracy, revealed a significant interaction between Cue Validity and Choice Accuracy (F(2,66175)=92.23, p<0.001), which we followed up with separate mixed effect analyses for

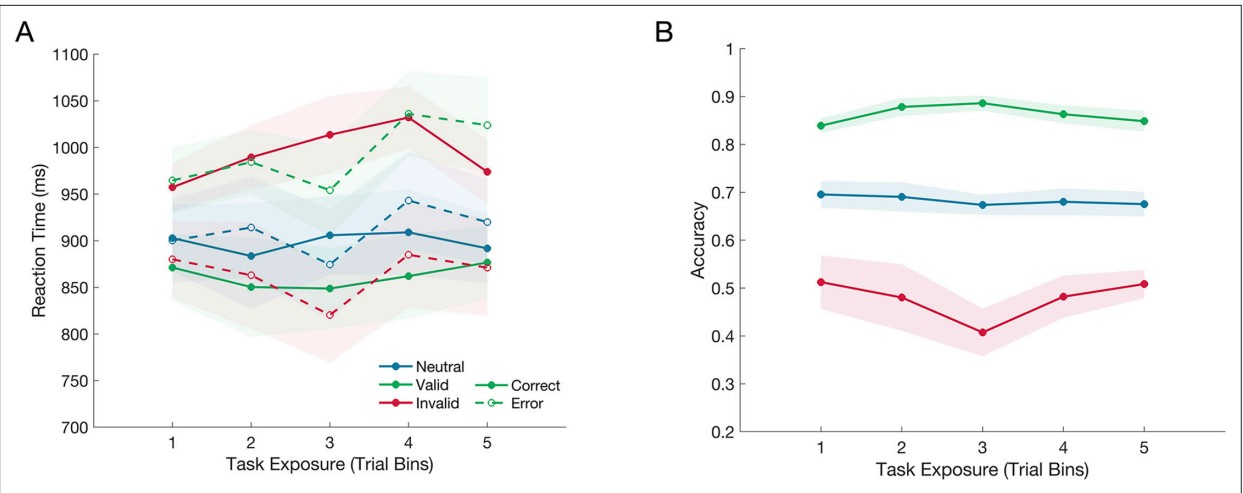

**Figure 2.** The effect of the cue on reaction time and choice accuracy over the course of testing, as indicated by the task exposure bins. (**A**) The cue significantly influenced reaction time for both correct and error responses, but this effect did not change over the course of testing. (**B**) The cue also significantly influenced accuracy, but again, this effect did not interact with the task exposure. The shaded regions represent the standard error of the mean (n = 12).

correct responses and error responses (**Figure 2**). The analysis of correct response reaction times revealed a main effect of the Cue Validity ($F_{(2,50776)}$=64.41, p<0.001) and reaction times significantly increased with Task Exposure ($F_{(1,50776)}$=17.92, p<0.001), but there was no interaction between the Cue Validity and Task Exposure ($F_{(2,50776)}$=1.48, p=0.227). Compared to neutral cue trials, reaction times were significantly faster on valid cue trials (p<0.001; $\Delta$=−0.043 ± 0.04) and significantly slower on invalid cue trials (p<0.001; $\Delta$=0.085 ± 0.06).

For error responses, there was also a main effect of the Cue Validity ($F_{(2,15386)}$=36.47, p<0.001) and reaction times also significantly increased with Task Exposure ($F_{(1,15386)}$=33.03, p<0.001), but again there was no interaction between the Cue Validity and Task Exposure ($F_{(2,15385)}$=0.15, p=0.865). Compared to the neutral cue condition, error reaction times were significantly slower following a valid cue (p<0.001; $\Delta$=0.084 ± 0.08) and significantly faster following an invalid cue (p<0.001; $\Delta$=−0.047 ± 0.08).

A mixed effects analysis of choice accuracy indicated that there was a significant effect of the Cue Validity ($F_{(2,66191)}$=692.46, p<0.001). However there was no significant effect of Task Exposure ($F_{(1,66191)}$=0.23, p=0.63), and no interaction between the Cue Validity and Task Exposure ($F_{(2,66191)}$=0.53, p=0.588). Compared to the neutral cue condition, participants' accuracy significantly increased when they were given a valid cue (p<0.001; $\Delta$=0.184 ± 0.01) and significantly decreased when they were given an invalid cue (p<0.001; $\Delta$=−0.219 ± 0.007).

The trend of increasing reaction times and stable accuracy is likely at least partly attributable to the titration of task difficulty across sessions. While accuracy was maintained across testing sessions, a regression analysis showed that the marginal contrast was significantly reduced as a function of task exposure ($F_{(1,58)}$ = 7.17, p=0.01, $R^2$=0.332). The mean marginal contrast across task exposure bins from first to last was 16.3% (SD = 6.4%), 13.08% (SD = 4.8%), 11.38% (SD = 4.1%), 11.5% (SD = 3.9%), 11.4% (SD = 3.7%). Despite the reduction in differential contrast across task exposure, task performance remained stable as participant scores did not change significantly across trial exposure bins ($F_{(1,58)}$ = 0.004, p=0.953).

## The effect of the cue on the encoding of sensory evidence

Our SSVEP analyses centred on the 'marginal SSVEP': the difference between the amplitude of the signals generated by the target and non-target gratings. As part of an experimental manipulation whose effects will be examined in a separate paper, we introduced brief (150 ms) evidence pulses on a subset of trials in which the contrast difference was transiently increased or decreased (see Materials and Methods). Unless otherwise specified, the marginal SSVEP was measured in the window 680–975ms post evidence onset because that interval falls after the final pulse offset time (650 ms).

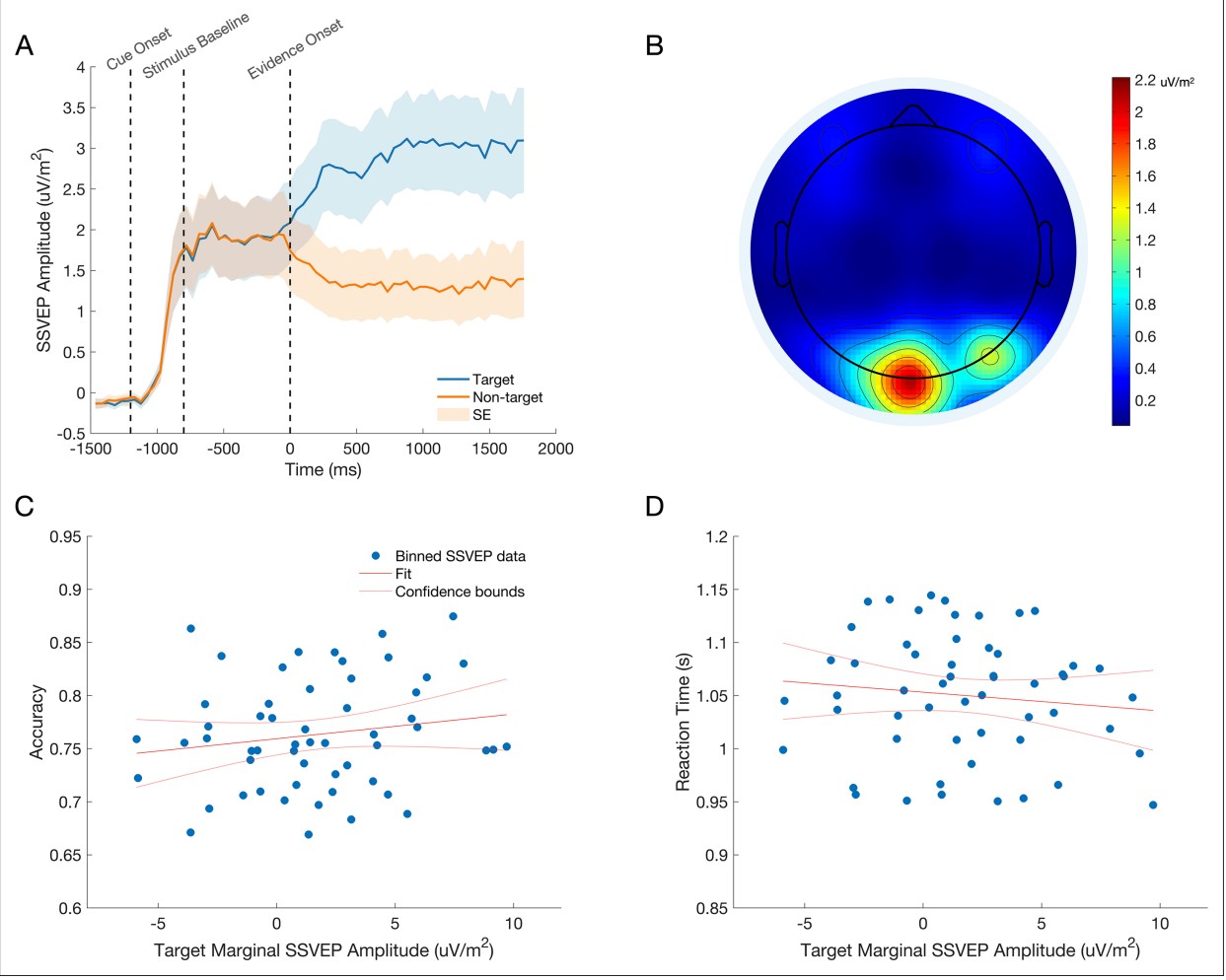

**Figure 3.** The SSVEP tracks stimulus contrast. (**A**) The dashed time markers represent the onset of the fade-in sequence, the onset of the baseline phase, and the onset of evidence. The mean SSVEP can be seen to rise as the stimulus fades-in. The signal then plateaus for the 800ms baseline phase, where both gratings are presented at 50% contrast. Finally, the signal clearly discriminates the target stimulus from the non-target stimulus as evidence onsets at 0ms. The shaded regions represent the standard error of the mean (n = 11). (**B**) The topography of the SSVEP signal during evidence presentation shows strong activity over visual cortex. (**C**, **D**) The relationship between marginal SSVEP amplitude and choice accuracy (**C**) and reaction time (**D**). Data points represent each subject's data divided into quintiles according to marginal SSVEP amplitude. For illustration, a linear regression trend line is shown with 95% confidence bounds in red in each plot.

Confirming that the SSVEP was a reliable neural index of the encoded contrast evidence, a Bayesian one-sample t-test indicated that the target (contrast-increase) SSVEP was significantly greater than the non-target (contrast-decrease) SSVEP (t(10) = 6.67, p<0.001, *Figure 3A*) in the neutral cue condition, with a Bayes Factor$_{10}$ of 500, which is considered very strong evidence.

We first sought to characterise the influence of the marginal SSVEP signal on the outcome of the decision process. We used a binary logistic mixed effects model to predict accuracy on a trial-by-trial basis, using the amplitude of the marginal SSVEP for trials with reaction times greater than 680ms (*Figure 3C*). To control for its effects on choice accuracy the cue validity was also included in the model. There was a significant main effect of the marginal SSVEP (F(1,31186)=5.183, p=0.023) and Cue Validity (F(2,31186)=1289.273, p<0.001). The model coefficient indicated that accuracy significantly increased as the amplitude of the marginal SSVEP increased ($\beta$=0.009, p=0.023; see *Figure 3C*). A second mixed effects analysis revealed a significant effect of the marginal SSVEP (F(1,31186)=7.319, p=0.07) and Cue Validity (F(2,31176)=45.784, p<0.001) on reaction time. The model coefficient indicated that reaction times significantly decreased as the amplitude of the marginal SSVEP increased ($\beta$=−0.001, p=0.007; see *Figure 3*.D).

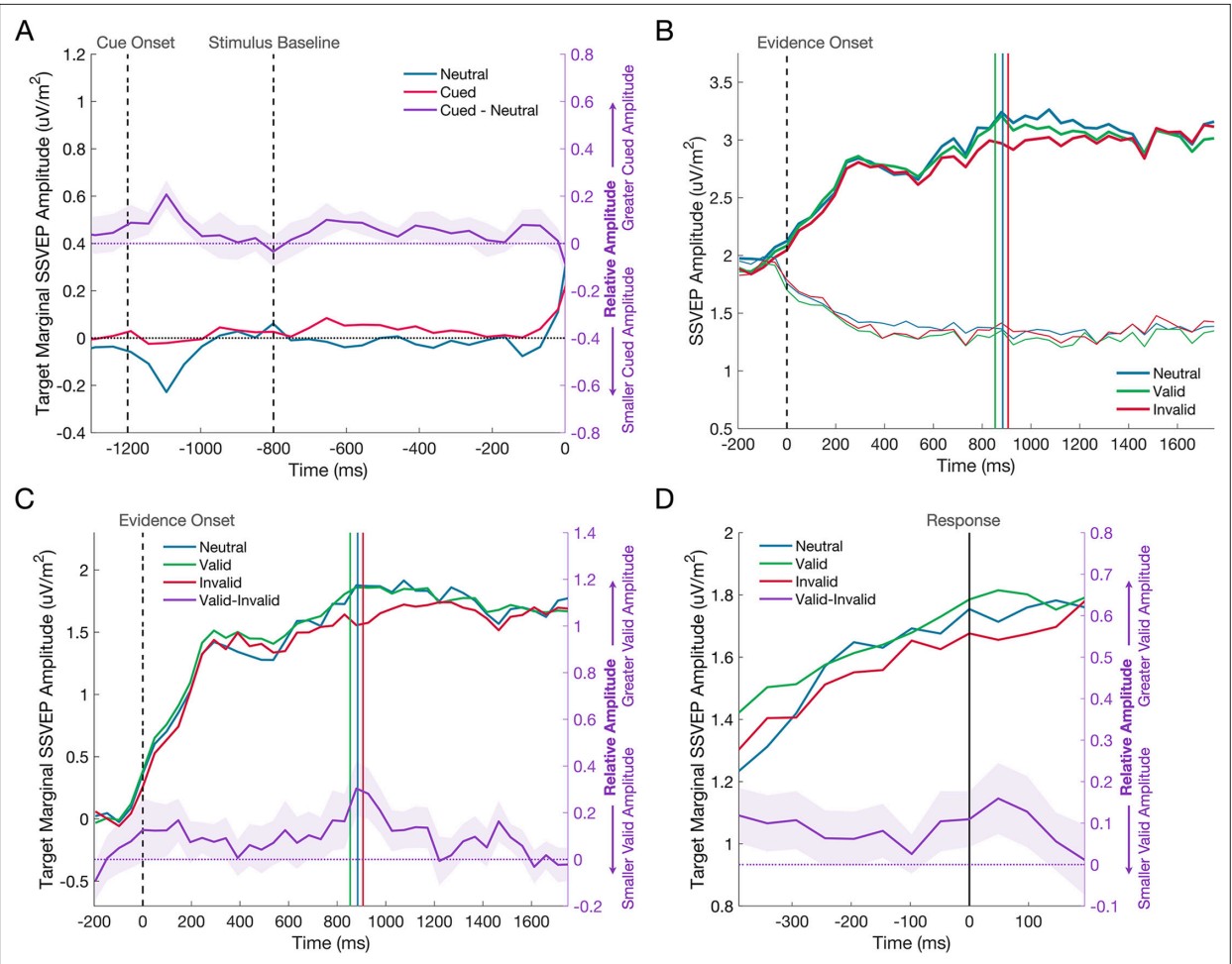

**Figure 4.** The SSVEP response to the predictive cue. (**A**) There was no significant difference in the amplitude of the SSVEP across cued and neutral cue trials during the baseline phase of the trial, when the cue had been shown, but the gratings were presented at equal contrast. The cue was presented at –1200ms, the grating stimuli reached 50% contrast at –800ms (marked by the first dashed vertical line), and evidence onset at 0ms (marked by the second dashed vertical line). The deflection at approximately –1100ms is likely a response to the presentation of the directional cue. The shaded regions represent the standard error of the mean (n = 11). The difference between neutral and cued conditions is shown on the right axis in purple. (**B**) The effect of the cue on the evidence-locked SSVEP is shown for the target (thick line) and non-target (thin line) signals separately. (**C**) The target marginal SSVEP in the invalid cue condition has a reduced amplitude compared to the neutral and valid cue conditions. The dashed vertical line represents evidence onset and the coloured vertical lines represent the median reaction times for each cue condition. The difference between valid and invalid cue conditions is shown on the right axis in purple. (**D**) The response-locked SSVEP across cue conditions, where the vertical line marks the response. The same trend of a reduced amplitude in the invalid cue condition can be seen at all time points leading to the response. Again, the difference between valid and invalid cue conditions is shown on the right axis in purple.

The online version of this article includes the following figure supplement(s) for figure 4:

**Figure supplement 1.** Reaction time quantiles across cue conditions.

Several studies have indicated that expectations can evoke preparatory sensory activity before evidence onset (e.g. *Kok et al., 2017*; *Trapp et al., 2016*), which may itself constitute an early source of sensory evidence fed into the decision process (*Feuerriegel et al., 2021a*). To investigate this hypothesis, we measured the SSVEP amplitude in the pre-stimulus, baseline phase of the trial (–752:–214ms); comparing the marginal SSVEP representation of the cued stimulus (i.e. cued minus uncued SSVEP) to the marginal representation of the upcoming target stimulus (i.e. target minus non-target) in the neutral cue condition (*Figure 4A*). The cues did not produce a significant anticipatory modulation of sensory encoding (Cued vs Neutral, F(1,49289)=1.31, p=0.253).

Addressing our primary hypothesis, a mixed effects analysis was conducted on the marginal SSVEP during evidence presentation to determine if the cue exerted any effect on the SSVEP representation

of the sensory evidence (see *Figure 4B, C*). The analysis showed a main effect of the Cue Validity on the marginal SSVEP (F(2,44553)=4.21, p=0.015). The marginal SSVEP was significantly reduced following an invalid cue compared to either a valid cue (p=0.004; $\Delta$=–0.148 ± 0.051) or a neutral cue (p=0.037; $\Delta$=–0.136 ± 0.065). There was no significant difference between valid and neutral cue conditions (p=0.807; $\Delta$=0.013 ± 0.051). A follow-up analysis on the amplitude of the target and non-target SSVEP was used to investigate whether the cue effect was primarily driven by a modulation of either or both of these signals. As expected, SSVEP amplitudes were significantly larger when elicited by Target compared to Non-Target gratings (F(1,89120)=4463.27, p<0.001) and the effect of the Cue Validity remained significant (F(2,89120)=3.48, p=0.031), but there was no Cue by Target/Non-Target interaction (F(2,89120)=2.92, p=0.054).

## The Influence of task exposure on motor- and sensory-level effects

Several studies have shown that motor preparation signals are adjusted in response to predictive cues such that starting levels of motor preparation for the cued effector are elevated (e.g. *de Lange et al., 2013*; *Kelly et al., 2021*). We examined cue effects on motor preparation to compare the time course of their emergence to the patterns observed in the SSVEP. Visual inspection of the waveforms in *Figure 5* suggests that there was strong lateralisation of motor signals during the baseline phase of each trial toward the expected alternative and this effect was evident in the earliest task exposure bin and remained largely stable across subsequent bins. This was confirmed by a mixed effects analysis of the baseline phase of the trial (–752:–215ms), which showed there was significantly greater MB lateralisation in favour of the expected alternative for cued compared to neutral trials (F(1,52540)=13.94, p<0.001), but no change in the extent of this lateralisation across Task Exposure (F(1,52540)=0.05, p=0.819) and no interaction between the Cue Condition and Task Exposure (F(1,52540)=0.23, p=0.631).

Having established in the previous section that cues did modulate SSVEP amplitude, we conducted a second analysis to determine if this effect changed over the course of testing (*Figure 5*). A mixed effects analysis revealed a significant decline in marginal SSVEP as a function of Task Exposure (F(1,44550)=69.16, p<0.001). We attribute this decline to the per-session titration of task difficulty which resulted in a progressive reduction in the marginal contrast of the grating stimuli. We also observed a significant interaction between Cue Validity and Task Exposure driven by the fact that the cue validity effect on the marginal SSVEP only emerges in the later stages of testing (F(2,44550)=4.31, p=0.013). The model coefficients indicated that there was a significantly greater decline in amplitude as a function of task exposure in the invalid cue condition ($\beta$=–0.123, p=0.007). With the model accounting for this interaction, there was no longer the main effect of Cue Validity (F(2,44550)=1.20, p=0.302) that was observed in the previous analysis. Thus, whereas starting-levels of motor preparation were biased by the cues at the earliest stages of testing, the sensory-level modulations only emerged after substantial task exposure.

To test the possibility that the observed SSVEP modulation arises primarily after choice commitment (e.g. reflecting confirmation bias; see *Figure 4—figure supplement 1*), a mixed effects analysis was conducted on the difference between the pre- (–200:–50ms) and post-response marginal SSVEP (50:200ms). This 'difference SSVEP' increased with Task Exposure (F(2,42605)=4.03, p=0.045), but there was no main effect of Cue Validity (F(2,42605)=2.103, p=0.122). Furthermore, there was no interaction between Cue Validity and Task Exposure (F(2,42605)=2.38, p=0.093). To quantify the evidence that the cue effect was not driven by changes in the signal after the response, we ran Bayesian one-way repeated measures ANOVAs on the marginal SSVEP comparing the difference across cue conditions before and after the response. If the cue effect only emerged after the response, we would expect the difference between invalid and neutral or invalid and valid cues to increase in the post-response window. There was no compelling evidence of an increase in the effect when comparing invalid to neutral ($BF_{10}$=1.58) or valid cues ($BF_{10}$=0.32). As can be seen in the response-locked plot (*Figure 4D*), the amplitude of the grand average valid SSVEP is consistently greater than that of the invalid SSVEP before and after the response, indicating that the effect was not a purely post-choice phenomenon.

## The effect of the cue on stimulus engagement

A priori, one might have expected that a valid cue should boost the marginal SSVEP compared to the neutral cue, but we observed no such effect, with amplitude instead significantly reduced following

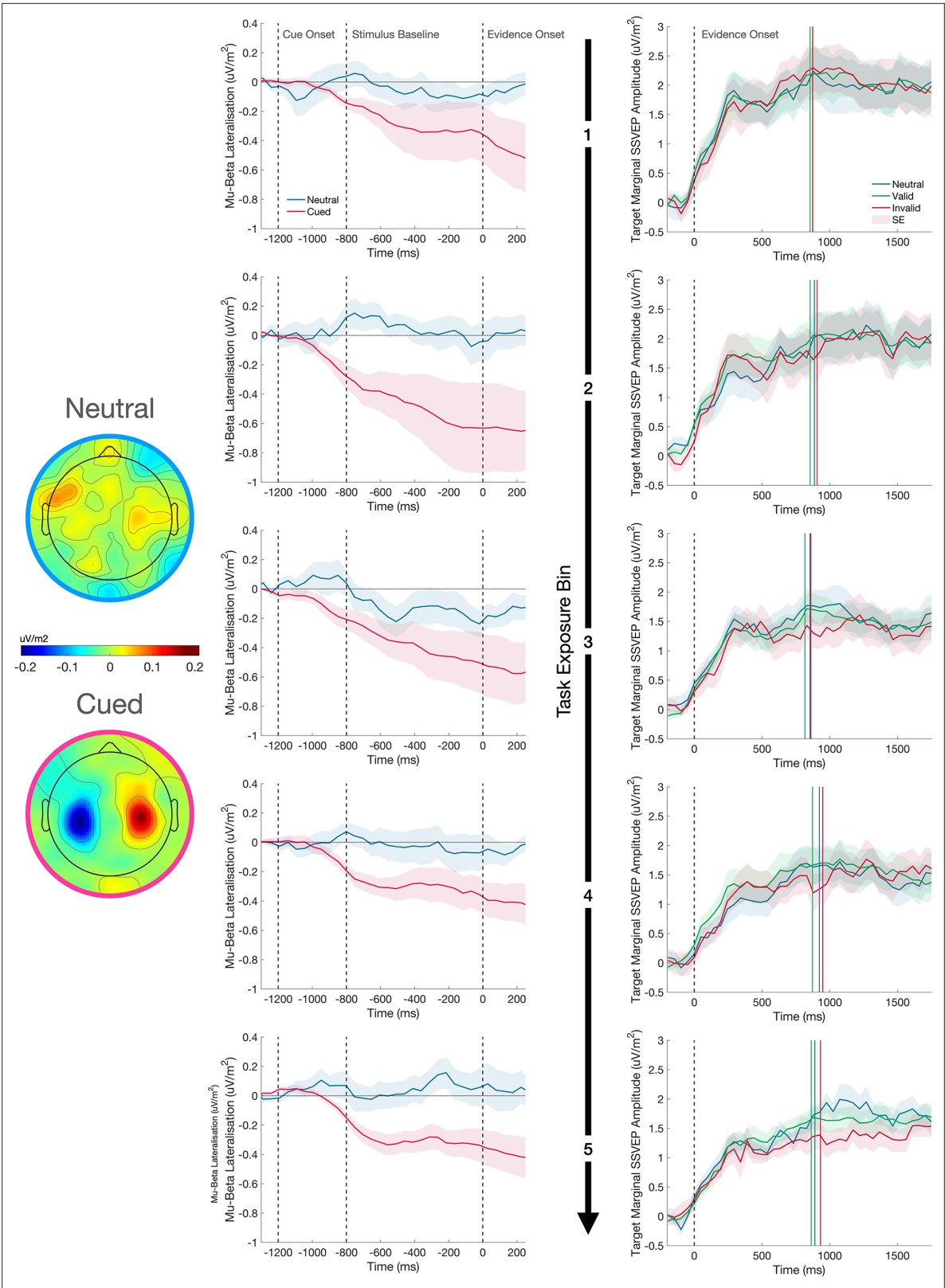

**Figure 5.** The effect of the cue on motor preparation before evidence onset, as indexed by mu-beta oscillatory activity (MB), and the representation of the sensory evidence (SSVEP) after evidence onset over the course of testing. MB lateralisation (left) and the target marginal SSVEP (right) are shown for each of the five task exposure bins separated by rows. In the MB plots, cue-lateralised activity (contralateral minus ipsilateral) is shown during the 800ms baseline phase of the trial after presentation of the cue and immediately prior to evidence onset (0ms), where each grating was held at 50%

*Figure 5 continued on next page*

*Figure 5 continued*

contrast. Greater lateralisation of MB activity provides a neural signature of the extent of motor preparation for the contralateral response. In the cued conditions, the signals are defined as contralateral and ipsilateral to the cue; in the neutral cue condition, the signals are defined as contralateral and ipsilateral to the correct response on that trial, which could not be known before evidence onset. The topography of MB activity in the window - 200:0ms before evidence onset is plotted on a common scale for neutral and cued conditions separately. There was no change in the degree of pre-evidence MB lateralisation across task exposure. In the SSVEP plots, evidence onset is marked by the dashed vertical line. The coloured vertical lines represent the median reaction times for each cue condition in each plot and the shaded regions represent the standard error of the mean (n = 12 for MB and n = 11 for SSVEP). The relatively reduced amplitude of the marginal SSVEP in the invalid cue condition emerges over the course of task exposure.

invalid cues relative to the other cue types. Given the high difficulty of the task (titrated to 70% accuracy for each session), we hypothesised that the presence/absence of cues may have impacted on the participants' degree of engagement with the stimulus. That is, in the absence of any predictive information on neutral trials, participants must rely entirely on their ability to efficiently accumulate samples of evidence to respond correctly, and therefore they may deploy greater attentional resources in order to optimise sensory encoding. It has been well established that alpha-band oscillatory activity over occipito-parietal sites desynchronises during focussed attention and predicts performance on perceptual discrimination and detection tasks (*Babiloni et al., 2006*; *Ergenoglu et al., 2004*; *Hansl-mayr et al., 2007*; *Kelly and O'Connell, 2013*; *Kelly et al., 2006*; *Kelly et al., 2009*; *Linkenkaer-Hansen et al., 2004*; *O'Connell et al., 2009*; *van Dijk et al., 2008*). This desynchronisation of alpha activity has been linked with activity in the dorsal attention network (*Laufs et al., 2003*; *Laufs et al., 2006*; *Mantini et al., 2007*; *Sadaghiani et al., 2010*; *Scheeringa et al., 2009*), and is thought to arise from the increased excitability in task-relevant processing regions and the suppression of task-irrelevant activity (*Jensen and Mazaheri, 2010*; *Romei et al., 2008*; *Thut and Miniussi, 2009*; *Van Diepen et al., 2019*). In a post-hoc analysis, we tested for greater task engagement in the neutral cue condition by measuring alpha-band oscillatory activity over a central occipito-parietal site in the pre-evidence window –410:–70ms (*Figure 6*). As with the MB analysis, the measurement window fell before evidence onset, allowing us to collapse the valid and invalid cue conditions into a single 'cued' condition. A mixed effect analysis showed a main effect of Cue Condition on alpha amplitude (F(1,52527)=10.10, *P*=0.001), such that there was significantly greater alpha desynchronisation in the neutral cue condition, consistent with increased task engagement in this condition.

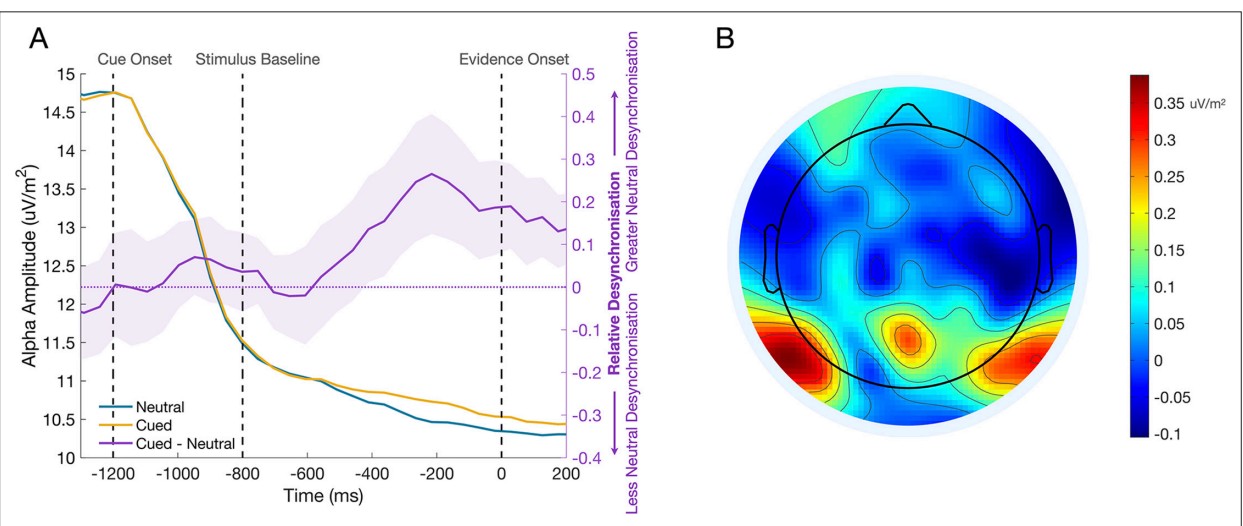

**Figure 6.** The effect of the cue stimulus engagement before evidence onset. (**A**) Alpha waveforms for the cued and neutral conditions are shown in yellow and blue on the left axis. The difference between these conditions (cued - neutral) is shown on the right axis, where the shaded region represents the standard error (n = 12). Alpha desynchronisation is greater in the neutral cue condition, suggesting that participants may approach neutral cue trials with greater attention to the stimulus to optimise their encoding of the evidence in the absence of predictive information. (**B**) The topography of the difference between cued and neutral conditions in the pre-evidence window –362:–215ms. The differential activity has the expected central occipito-parietal topography.

## Discussion

In the past decade, there has been a surge of interest in the role of expectation in perceptual processing, which has developed into a substantial literature reporting expectation-based modulations of neural activity across a variety of brain areas, species, paradigms, and recording techniques. This work has, in turn, spurred debate about the traditional characterisation of early sensory processing as insulated from cognitive influences. While some studies have presented evidence that probabilistic expectations induce behavioural biases without changing perceptual sensitivity (e.g. *Bang and Rahnev, 2017*; *Rao et al., 2012*; *Rungratsameetaweemana et al., 2018*), others have demonstrated anticipatory and/or stimulus-evoked sensory-level modulations (e.g. *Aitken et al., 2020*; *Albright, 2012*; *Egner et al., 2010*; *Ekman et al., 2017*; *Esterman and Yantis, 2010*; *Kok et al., 2012*; *Kok et al., 2013*; *Kok et al., 2014*; *Kok et al., 2017*; *Puri et al., 2009*; *Rahnev et al., 2011*; *Richter et al., 2018*; *Trapp et al., 2016*). However, this previous work either measured sensory signals whose relevance to the decision process is uncertain and/or used cues that predicted a forthcoming stimulus but not the correct choice alternative. To assess whether prior probabilities modulate sensory-level signals directly related to participants' perceptual decisions, we implemented a task in which the cues explicitly predicted the correct choice and in which sensory signals that selectively trace the evidence feeding the decision process could be measured during the process of deliberation. Like previous studies (*O'Connell et al., 2012*; *Steinemann et al., 2018*), we found that the SSVEP closely tracked the differential contrast representing the sensory evidence in this task and was a significant predictor of choice accuracy and reaction time, as one would expect of an evidence encoding signal. We also found that the differential SSVEP responses were significantly modulated by predictive cues consistent with the proposal that prior probabilities are capable of modulating neural activity at early cortical stages of perceptual processing even when those expectations are specifically about the sensory features being decided upon. Crucially, however, this effect only emerged with substantial task exposure, suggesting that previous studies may have overlooked such sensory-level effects because they administered significantly fewer trials (e.g. *den Ouden et al., 2023*).

Several studies suggest that expectations may lead to the generation of anticipatory feature-specific sensory templates, which shape the processing of the subsequent stimulus (e.g. *Blom et al., 2020*; *Ekman et al., 2017*; *Esterman and Yantis, 2010*; *Kok et al., 2014*; *Kok et al., 2017*; *Puri et al., 2009*; *Trapp et al., 2016*). Here, we did not observe any significant anticipatory expectational modulations of sensory activity when measuring SSVEP amplitudes during a baseline interval between cue and evidence onset, during which the overlaid grating stimuli were shown at equal contrast. However, typically, studies that have detected preparatory activity did not present a baseline version of the expected stimulus in the window where the anticipatory modulation was recorded (e.g. *Kok et al., 2014*; *Kok et al., 2017*; *Trapp et al., 2016*) and the preparatory activity has been reported to be significantly weaker than the responses to physical stimuli. It is possible that this relatively subtle activity was obscured in the present study by the response to the presentation of physical grating stimuli during the pre-evidence baseline period. It is also not clear how far in advance of stimulus onset this kind of preparatory modulation can be expected to manifest. To our knowledge, the only estimate of this was offered by *Kok et al., 2017* who found that the pre-stimulus activity only became predictive of the upcoming stimulus 40ms before stimulus onset. If that narrow window of predictive activity is representative of anticipatory modulations in general, the temporal resolution of the SSVEP would be insufficient to appropriately interrogate such an effect (as it relied on a 400ms window of activity). It is also plausible that our participants were sufficiently well-trained on the task timings that the influence of the prior was restricted to the period of evidence presentation without shaping the baseline phase where participants knew that the gratings were presented at equal contrast and had no reason to 'expect' otherwise. Indeed, some of the studies reporting anticipatory modulations used unpredictable stimulus onset times (e.g. *Trapp et al., 2016*) and there is some evidence that when participants are familiar with task timings, preparatory activity is initiated only when the predicted stimulus is expected to onset (e.g. *Esterman and Yantis, 2010*).

In contrast to the pre-evidence analysis, the cues did significantly impact on the marginal amplitude of the SSVEP following evidence onset, with significantly smaller responses following invalid cues compared to valid and neutral cue trials. This pattern is apparently at odds with a common observation in the expectation literature, that evoked responses to unexpected stimuli tend to be larger than responses to expected stimuli (*Feuerriegel et al., 2021b*; *Walsh et al., 2020*). These

expectation suppression effects have been interpreted as consistent with predictive processing schemes, where unexpected sensory events provoke prediction error signals that call for a revision of perceptual hypotheses (*Friston, 2005*). These error signals are thought to emanate from the same superficial pyramidal cells that are primarily responsible for generating EEG signals (*Cohen, 2017*; *Friston, 2009*), further underlining the apparent discrepancy between these schemes and the present SSVEP results. An alternative account for expectation suppression effects, which is consistent with our SSVEP results, is that they arise, not from a suppression of expected activity, but from a 'sharpening' effect whereby the response of neurons that are tuned to the expected feature are enhanced while the responses of neurons tuned to unexpected features are suppressed (*de Lange et al., 2018*). On this account, the expectation suppression commonly reported in fMRI studies arises because voxels contain intermingled populations with diverse stimulus preferences and the populations tuned to the unexpected features outnumber those tuned to the expected feature. In contrast to these fMRI data, the SSVEP represents the activity of sensory units driven at the same frequency as the stimulus, and thus better isolates the feature-specific populations encoding the task-relevant sensory evidence. Therefore, according to the sharpening account, an invalid cue would have enhanced the SSVEP signal associated with the low-contrast grating and weakened the SSVEP signal associated with the high-contrast grating. As this would result in a smaller difference between these signals, and therefore, a lower amplitude marginal SSVEP compared to the neutral cue condition, this could explain the effect we observed.

It would also be expected that a valid cue should have the converse effect, producing a stronger marginal SSVEP compared to the neutral cue condition. This pattern was not borne out in the data. However, there is reason to believe that the cued and neutral conditions do not only differ in terms of the predictive information conferred to the participant. Unlike in the cued conditions, participants were given no prior indication of the relative probability of the upcoming target in the neutral cue condition, meaning that they were required to rely solely on careful accumulation of the presented evidence with no preparatory strategic adjustments on a demanding task, designed to elicit only 70% accuracy. Indeed, analysis of pre-evidence alpha activity over occipito-parietal sites provided evidence of enhanced stimulus engagement in the neutral cue condition compared to the cued conditions. Variations in pre-stimulus alpha-band power have been shown to correlate with fluctuations in performance in a variety of perceptual tasks (e.g. *Babiloni et al., 2006*; *Ergenoglu et al., 2004*; *Hanslmayr et al., 2007*; *Linkenkaer-Hansen et al., 2004*; *O'Connell et al., 2009*; *van Dijk et al., 2008*), a phenomenon thought to arise from the preferential encoding of task-relevant information (*Jensen and Mazaheri, 2010*; *Van Diepen et al., 2019*). Together, this suggests that participants may have been more closely engaged with the stimulus on neutral cue trials and, as attention has previously been shown to boost the SSVEP (e.g. *Morgan et al., 1996*; *Müller et al., 2006*), this may explain the similar SSVEP waveforms in the valid and neutral cue conditions. This also raises the possibility that a valid cue may actually boost the SSVEP rather than, or as well as, the invalid cue suppressing the SSVEP since it appears that participants were able to achieve the same marginal SSVEP amplitude on valid as neutral cue trials with less effort/attention. This could be investigated in future research by using cues with scaled predictive probabilities (e.g. 60%, 70%, and 80%) made explicit to the participant. The direction of the modulation could be characterised by assessing changes in the response of both the target and non-target SSVEPs following cues with different probabilities.

Modelling studies have indicated that starting-point biases are the optimal way to incorporate prior knowledge into the decision process (*Bogacz et al., 2006*; *van Ravenzwaaij et al., 2012*) and are sufficient to account for prior probability effects on behaviour without the need to invoke modulations of the evidence itself (*Leite and Ratcliff, 2011*; *Mulder et al., 2012*; *Ratcliff and McKoon, 2008*). However, the emphasis on parsimony in formal model comparisons has tended to promote models that can capture experimental manipulations of behaviour with a single dominant parameter adjustment (*Purcell and Palmeri, 2017*; *Voss et al., 2004*), like a starting-point bias. Our model-free neurophysiological analyses reveal that the strong biases in motor preparation, which participants exhibited early on in testing, were accompanied by more subtle sensory encoding modulations, which only emerged with extensive task exposure (circa 4 hr). Participants' scores remained constant across task exposure, even as difficulty titration reduced the available physical evidence. So where one might have predicted that participants would learn to decrease their reliance on suboptimal strategies as task performance improves, our results point to the opposite trend. We suggest that participants may

have been able to stabilise their performance across task exposure, despite reductions in the available sensory evidence, by incorporating the small sensory modulation we detected in the SSVEP. This would suggest that the decision process may not operate precisely as the models used in theoretical work describe. Instead, our study tentatively supports a small number of modelling investigations that have challenged the solitary role of starting point bias, implicating a drift bias (i.e. a modulation of the evidence before or upon entry to the decision variable) as an additional source of prior probability effects in perceptual decisions (*Dunovan et al., 2014*; *Hanks et al., 2011*; *Kelly et al., 2021*; *van Ravenzwaaij et al., 2012*; *Wyart et al., 2012*) and indicates that these drift biases could, at least partly, originate at the sensory level. However, this link could only be firmly established with modelling in a future study.

It has been suggested that sensory-level effects may reflect attentional processes or post-decisional relaying of the decision state to sensory regions, without fundamentally changing the encoding of the sensory evidence (*Simon et al., 2019*). Indeed, the dissociation of expectation and attention has been a Gordian knot in expectation research, particularly due to the prevalence of cueing paradigms (*Aitchison and Lengyel, 2017*; *Alink and Blank, 2021*; *Rungratsameetaweemana and Serences, 2019*; *Summerfield and Egner, 2009*). Several authors have pointed out that when prior probabilities are manipulated using cues, subjects are provided with both probabilistic information about what is likely to appear and information about the relevance of stimulus features for the task being performed, conflating expectation and attention (*Bang and Rahnev, 2017*; *Rungratsameetawee-mana and Serences, 2019*; *Simon et al., 2019*). Indeed, shifts in feature-based attention have been found to induce feature-specific modulations of stimulus-evoked BOLD activity in early visual areas (e.g. *Kamitani and Tong, 2005*; *Serences and Boynton, 2007*). An advantage of the use of overlaid grating stimuli in the present paradigm was that there was no obvious advantage to shifting attention to a particular region of space to better identify one of the stimulus alternatives. In addition, both gratings were equally relevant to the choice, regardless of the correct alternative since participants were asked to compare their relative contrast. Nevertheless, the possibility that participants may have preferentially attended to the grating that was expected to appear at higher contrast cannot be excluded here. If an invalid cue led participants to pay closer attention to the non-target orientation, this may have degraded the differential representation of the contrast evidence. It is possible that the reason attention did not significantly enhance the marginal SSVEP when valid cues were presented was because the physical evidence had already guided attention to the target grating on neutral and valid cue trials. However, even if the SSVEP modulations reported here arise from feature-based attention rather than a sensory-prior, they would still constitute a modulation of the sensory evidence feeding the decision process based on prior probabilities.

An implication of using cues that predict not just the upcoming stimulus, but the most likely response, is that it becomes difficult to determine if the preparatory shifts in mu-beta (MB) activity that we observed reflect adjustments directly influencing the perceptual interpretation of the stimulus or simply preparation of the more probable action. When perceptual decisions are explicitly tied to particular modes of response, the decision state can be read from activity in motor regions associated with the preparation of that kind of action (e.g. *de Lafuente et al., 2015*; *Ding and Gold, 2012*; *Shadlen and Newsome, 2001*; *Romo et al., 2004*), but these modules appear to be part of a constellation of decision-related areas that are flexibly recruited based on the response modality (e.g. *Filimon et al., 2013*). When participants cannot prepare a response in advance or no response is required, MB no longer traces decision formation (*Twomey et al., 2016*), but an abstract decision process is still readily detectable (e.g. *O'Connell et al., 2012*), and modelling work suggests that drift biases and starting point biases continue to influence prior-informed decision making (*Thomas et al., 2022*; *Yon et al., 2021*). While the design of the present study does not allow us to offer further insight about whether the MB effects we observed were inherited from strategic adjustments at this abstract level of the decision process, we hope to conduct investigations in the future that better dissect the distinct components of prior-informed decisions to address this question.

Several other issues remain unaddressed by the present study. It is not clear to what extent the sensory effects may be influenced by features of the task design (e.g. speeded responses under a strict deadline) and if these sensory effects would generalise to many kinds of perceptual decision-making tasks or whether they are particular to contrast discrimination. This is an important area for further research as some of the studies that have reported no evidence of sensory modulations by expectation

have used motion discrimination (*Rao et al., 2012*), orientation discrimination (*Rungratsameetawee-mana et al., 2018*), or expanded judgement tasks (*Bang and Rahnev, 2017*). Additionally, in the present study, a pulse of evidence could be presented within the first 650 ms of a trial. To minimise contamination of the SSVEP dynamics with these evidence pulses, the effect of the cue was measured in the window 680–975ms, so it is not clear whether the cue also influenced the SSVEP in the earliest stages of evidence presentation. Some have suggested that expectation-based modulations may only influence the encoding of a stimulus after an initial volley of sensory processing (*Aitken et al., 2020*; *Alilović et al., 2019*; *Press et al., 2020*). Although the temporal resolution of the SSVEP is limited by the window of activity needed for Fourier analysis, future work could provide greater insight on when the cue effect emerged by removing the pulses from the paradigm. While our goal was to assess gradual trends in the emergence of these effects with task exposure, future studies could also investigate the possibility that this process is not linear. For example, participants may respond to the emergence of a sensory modulation with discrete strategic adjustments.

In conclusion, we used a contrast-sensitive electrophysiological signature of sensory activity to determine if prior probability exerted an influence on the encoding of sensory evidence feeding the decision process. Probabilistic cues were found to modulate the amplitude of the SSVEP, such that the representation of the marginal contrast of unexpected stimuli was relatively diminished. In addition, the effect only emerged over the course of prolonged exposure to the task. A modulation of the original encoding of the sensory evidence would have knock-on effects at each stage of the decision process. This suggests that, in addition to the preparatory strategic adjustments that are characteristic of expectation-based decision-making, prior probability may also be instantiated in the dynamics of the ongoing deliberation. This is consistent with recent work which implicates the subtle orchestration of several distinct parameter adjustments in prior-informed decisions (*Kelly et al., 2021*). We suggest that this effect may represent a candidate mechanism for a sensory-level contribution to this dynamic.

## Materials and methods

### Participants

Twelve adults participated in this study (five female, age range: 18–39, M=25.2 ± 6.7). However, a reliable SSVEP could not be established for one subject, so they were excluded from the SSVEP analyses (n=11), but were retained for all other analyses (n=12). All participants reported normal or corrected-to-normal vision, no history of migraine or bad headaches, no history of epilepsy, and no sensitivity to flashing light. These exclusion criteria were established prior to recruitment. Participants provided informed written consent prior to testing and were paid a gratuity of €10 per hour of participation and an additional €20 upon completion of all sessions as compensation for their time. All procedures were approved by the Trinity College Dublin School of Psychology Ethics Committee (ref SPREC042020-15) and were in accordance with the Declaration of Helsinki. Participants completed between 4 and 6 testing sessions, each on a different day. While the sample size was small, on average, participants completed 5750 (SD = 1066) trials each. This small-N, high trial count approach was chosen to minimise measurement error associated with the noisy EEG data in an effort to detect relatively subtle within-subject effects (*Baker et al., 2021*; *Smith and Little, 2018*).

### Experimental design

The experiment was conducted in dark, sound-attenuated testing booths. Visual stimuli, generated using Psychtoolbox (*Kleiner et al., 2007*) and a custom MATLAB experimental script (available in OSF repository), were presented on one of two displays, either a 51 cm or a 40.5 cm gamma-corrected CRT monitor (both monitors had a 1024x768 resolution and 100 Hz frame rate). A chin rest was used to reduce head movement and ensure that the viewing distance was 60 cm.

Participants completed a two-alternative forced-choice contrast discrimination task, where they decided which of two orthogonally-oriented overlaid gratings was presented with a higher contrast on discrete trials (see *Figure 1*). The stimuli were square-wave gratings with a spatial frequency of 1 cycle per degree of visual angle. Each grating was tilted by 45° relative to the vertical midline (one right tilt and one left tilt). The gratings were annular with an inner radius of 0.3° visual angle and an outer radius of 4° of visual angle, presented centrally on a grey background with the same mean luminance. The divergences of the contrast levels from the baseline level of 50% were reciprocal across gratings,

so if the target grating was set to 60% contrast, the other grating would be set to 40% contrast. The differential contrast was set during a calibration phase at the beginning of the session (see Procedure section). The starting phase of the gratings was shifted a half-cycle on every trial to reduce the potential influence of adaptation. The stimulus was designed to evoke an SSVEP to provide an independent measurement of the neural representation of the sensory evidence for each of the grating stimuli. This was achieved by separately 'frequency-tagging' the gratings by reversing their spatial phase at 20 and 25 Hz. The assignment of phase reversal frequency was randomly counterbalanced between the gratings across trials. A yellow fixation point, with a radius of 0.3° of visual angle was presented in the central annulus to reduce eye movements.

Trials began with a 1000 ms fixation period, where only the fixation point was displayed. To avoid visual evoked potentials, this was followed by a 400ms fade-in sequence where both gratings gradually increased from 0% to 50% contrast. The fade-in was followed by an 800ms baseline period, where the gratings maintained a 50% contrast level before the evidence was presented. Evidence onset involved a reciprocal change in the contrast of the target and non-target gratings by an amount determined by a QUEST procedure per individual (described below; *Watson and Pelli, 1983*). The target stimulus was shown for 2000 ms and the response window was 1800 ms. The stimulus presentation extended past the response window to facilitate time-frequency analyses requiring a 400ms window (see Time-Frequency Analyses for further details). A feedback screen was then shown for 1000ms.

Each trial contained one of three predictive cues, which were presented for 300 ms during the fade-in sequence at the beginning of each trial. Valid cues correctly identified the tilt of the target stimulus on the subsequent trial; invalid cues indicated the opposite of the target tilt; and neutral cues provided no information about the likely answer. Valid cues were presented four times as frequently as invalid cues, making them 80% predictive; invalid and neutral cues were presented on the same number of trials in each session. The cue was always a double-sided arrow, half-yellow and half-blue, with the yellow side indicating the cue direction. In the neutral cue condition, both sides of the arrow were yellow. The order of cue presentation was randomised.

The task also incorporated brief 'pulses' of evidence on some trials. However, the analyses presented in this paper did not investigate the influence of the pulses, so they are only described here for completeness. On no-pulse trials, the contrast difference between the correct and incorrect grating was constant across the evidence presentation. Pulse trials contained one of three types of evidence pulses: (1) reverse pulses flipped the evidence to favour the opposite grating. For example, if the contrast level of the target grating was 60% (and therefore, the contrast level of the non-target grating was 40%), during the reverse pulse, the non-target grating's contrast would increase to 60% and the target grating's contrast would fall to 40%; (2) a gap pulse eliminated all evidence for the duration of the pulse (i.e. the contrast of both gratings was set to 50%); and (3) positive pulses increased the evidence for the target grating. The magnitude of this increase was designed to be equivalent to the reverse pulse, so the differential evidence for the correct grating was added to the correct grating contrast and subtracted from the incorrect grating contrast. If the contrast level of the target grating was 60%, the total contrast change during a reverse pulse was 40% (60 ➞ 40% and 40 ➞ 60%) and the total contrast change during a positive pulse was also 40% (60 ➞ 80% and 40 ➞ 20%). The contrast values of the gratings were always reciprocal, ensuring that the pulses were not associated with a change in the average contrast. The pulses were presented for 150ms with one of five onset times beginning at 180ms and staggered in 80ms increments up to 500ms into the evidence presentation; their onset and offset were each instantaneous. The pulses were staggered to prevent a predictable onset being incorporated into a subject's response strategy and most subjects reported no awareness of the pulses when questioned at the end of testing. Each of the pulse conditions were equally represented in each session, meaning ~94% of trials contained pulses.

If the subject responded before the evidence onset, an exact replica of that trial was added to the end of the current block. An extra block was also included at the end of each session to re-present any pulse trials where the subject responded before the pulse was presented. The extra block was initially empty, however when any response occurred before the pulse onset, a set of three trials were added. One of these trials was a replica of the current trial, the remaining two trials had a different cue type and pulse-onset combination and were randomly chosen from the remaining alternative conditions. The replica trial was marked as immutable, but the other trials were free to vary if needed. If there was another response before pulse onset on a subsequent trial, a replica of that trial was swapped for one

of the mutable trials in the extra block. If there were no more mutable trials or more than 20% of trials in the extra block had the same pulse type, a new set of three trials was added. If either replacement scenario occurred while the subject was currently progressing through the extra block, the replica trial was just added to the end of the block.

## Procedure

For all sessions, the participant was comfortably seated in the testing booth and were instructed to maintain fixation on the centrally presented fixation point throughout all tasks. Subjects responded on each trial using a mouse held in the palm of their hands with their thumbs resting on each mouse button. To indicate the right-tilted grating had the stronger contrast, the participant pressed the right button with their right thumb and to indicate the left-tilted grating had the stronger contrast, they pressed the left button with their left thumb.

Each session began with a calibration phase to individually titrate the contrast differential between target and non-target stimuli to achieve 70% accuracy. The first calibration session was composed of five stages; subsequent calibrations comprised fewer stages and were designed to allow for adjustment of these initial calibration values if subject performance changed across sessions (e.g. due to practice effects). All calibration tasks used the same overlaid gratings stimulus described in the previous section.

The participant was introduced to the task by completing a 50-trial practice block. This practice block was mostly identical to the task schematic shown in *Figure 1*, but there were no cues or evidence pulses. Subjects were told that they had 1.8 s to respond once the contrast changed from the baseline period (both gratings 50% contrast) and to familiarise them with the timing of the task, this response window would be indicated by a change in the colour of the fixation point from yellow to blue. The first trial presented the target grating at 100% contrast and the other grating at 0% contrast (i.e. 100% evidence). The contrast differential decreased in small increments each time the subject gave two consecutive correct responses and increased if they gave two consecutive error responses. Subjects received feedback on each trial. The practice block was only given on the first day of testing. The timing cue was not included in any subsequent task.

Previous experiments using the frequency-tagging technique indicated that the grating that is modulated at the higher frequency is perceived as being presented at lower contrast even with identical physical contrast (e.g. *Devine et al., 2019*; *Kim et al., 2007*). Two calibration tasks were designed to address this issue. In the first of these tasks, participants completed two interleaved one-up-two-down staircase procedures to estimate the contrast difference that would achieve ~70% accuracy when both of the gratings were modulated by the same frequency (i.e. both 20 Hz or both 25 Hz). The differential evidence started at 100% and was adjusted by the staircase procedure according to the participant responses. The staircase ended after four reversals or after 50 trials. By comparing the contrast difference produced by these staircases, the perceptual bias could be estimated, and a contrast boost could be used to compensate for the perceptual flicker effect.

The second of these tasks was designed to validate this compensatory contrast boost using a second, one-up-one-down staircase. In this task, the 25 Hz grating was always the correct answer, and the 20 Hz grating was always incorrect. The staircase was designed to indicate the contrast boost to the 25 Hz grating required to achieve 50% accuracy (i.e. to make the two gratings indistinguishable), when each grating was modulated by a different frequency. The gratings were each set to 50% contrast and the 25 Hz grating was given the compensatory contrast boost estimated using the first staircase. If there was no perceptual bias, the subject would achieve 50% accuracy. The staircase ended after four reversals or 60 trials. If there was an indication of a systematic bias, the contrast boost was implemented in all subsequent tasks. However, in practice, only four participants showed a reliable bias in the first calibration session and of those, only one continued to demonstrate this bias when these tasks were rerun in future calibration sessions.

After the flicker bias had been estimated, the participant completed a QUEST procedure (*Watson and Pelli, 1983*) to estimate the contrast differential (i.e. difficulty level) required to achieve 70% accuracy on the task. The procedure was initially fed the final contrast differential achieved in the practice block and then dynamically updated this estimate over 60 trials. If the previous tasks indicated that there was a reliable perceptual bias based on flicker frequency, the stimulus incorporated the estimated compensatory boost to the 25 Hz stimulus. If the QUEST procedure estimated that

a subject required the target stimulus to have a contrast level greater than 65%, this was deemed an indication of insufficient practice, and the subject was asked to complete another practice block before repeating the QUEST procedure.

Finally, subjects completed 30 trials with this QUEST-estimated contrast differential to verify that it would result in 70% accuracy. If performance differed from 70% by less than 5%, the researcher adjusted the contrast differential based on the margin of error provided by the QUEST estimate and proceeded to the main task. If performance differed from 70% by more than 5%, the contrast differential was adjusted based on the QUEST margin of error and the verification block was rerun. The success of this titration procedure was verified with trials from the neutral cue condition from the main experiment, where the grand average accuracy was ~70%.

Once the calibration phase was completed, the subjects started the main task. To accommodate the participants, there was a long (~1152 trials) and a short (~768 trials) version of the testing session. The trial numbers are approximate because they were subject to increases as the trial replacement mechanism was triggered. During the main task, breaks occurred after every 48 trials, when a 20 s timeout was enforced. Participants were told that, although they were not always accurate, the cues predicted the correct answer and they should pay close attention to them, registering the direction of the cue on every trial. The experimenter explained that they should respond as soon as they were quite sure of their decision and try to maximise their score on each block. If their response was correct, the fixation point turned green and subjects were shown that they had received 50 points; if the response was an error, the fixation dot turned red, and subjects were shown that they had lost 25 points. If the subject responded before the target onset or if they failed to respond within 1800ms, the fixation point turned red, they were informed that they had lost 25 points and had been too fast or too slow in their response, respectively. Finally, the participants were not informed that there were pulses of evidence in the task. The experimenter explained that the stimulus was flickering to help with the analysis of the neural signals and that they could ignore these rapid changes. The pulses were difficult to detect amongst the gratings' frequency-tagged modulations, even when one was aware that there were pulses in the task design.

## EEG acquisition and preprocessing

Continuous EEG data were acquired from 128 electrodes using a BioSemi ActiveTwo system and digitised at 512 Hz. Eye movements were recorded using two vertical electrooculogram (VEOG) electrodes placed above and below the left eye. The data were analysed with custom scripts in MATLAB using the EEGLAB toolbox (*Delorme and Makeig, 2004*). The EEG data were detrended and low-pass filtered below 40 Hz. Channels identified as uniquely noisy using a custom channel variance analysis were recorded for each subject and each recording session and interpolated using spherical splines. The EEG data were then re-referenced offline using the average reference. The data were segmented into a cue-locked epoch (–1700:500ms relative to evidence onset) and an evidence-locked epoch (–400:2000ms relative to evidence onset). The cue-locked epoch was baseline corrected in the interval –1400:–1200ms and the evidence-locked epoch was baseline corrected in the interval –600:–400ms, both relative to evidence onset. Artifact rejection was conducted separately for the cue-locked and evidence-locked epochs. If the difference in activity between the VEOG channels exceeded an absolute value of 250 μV or if the voltage recorded by any scalp electrode exceeded 100 μV at any time during the epoch, that trial was excluded. After artifact rejection, a response-locked epoch (–600:400ms relative to the response) was created from the evidence-locked data. To compensate for the effects of volume conduction across the scalp, each epoch was subjected to a Current Source Density (CSD) transformation (*Kayser and Tenke, 2015*). This technique is used to minimise the spatial overlap between functionally distinct EEG components (*Kelly and O'Connell, 2013*).

The neurophysiological influence of the cue was measured with electrophysiological signals associated with sensory encoding and motor preparation. The SSVEP is evoked over occipital electrodes by flickering the stimulus at a specific frequency. The amplitude of this oscillatory signal scales with increases in stimulus contrast, so it is commonly used as an electrophysiological marker of the sensory encoding of contrast stimuli (*Norcia et al., 2015*; *O'Connell et al., 2012*; *Steinemann et al., 2018*). SSVEP electrodes were chosen from an occipital candidate pool. Electrodes were ranked by the difference in activity associated with the target and non-target stimulus (i.e. their contrast discrimination) in the window 200:1800ms after evidence onset. The top two electrodes were chosen for each subject.

In addition, mu-beta (MB) oscillatory activity contralateral and ipsilateral to the response provides a distinct read-out of the motor preparation for each response alternative (*de Lange et al., 2013*; *Donner et al., 2009*). Left and right hemisphere MB electrodes were separately selected for each participant from a pool of central electrodes. Electrodes were ranked based on two criteria: (1) the difference in amplitude at response when the candidate electrode was contralateral and ipsilateral to the response; (2) the slope of activity at the candidate electrode in the 400ms preceding the contralateral response. The top two ranked electrodes were chosen for each hemisphere for each participant. The same three occipito-parietal midline electrodes were selected for all subjects in the analysis of alpha activity.

## Time-frequency analyses

The Short Time Fourier Transform (STFT) procedure was used to decompose the EEG recording of neural activity into its time-frequency components to extract alpha (8–14 Hz), MB (8–30 Hz) and SSVEP signals associated with the tilted gratings (20 Hz and 25 Hz). The STFT window for MB and SSVEP analyses was 400ms, which was chosen to accommodate 10 cycles of the 25 Hz SSVEP signal. The STFT window was 360ms in the analysis of alpha activity. The windows were moved along the length of the epoch in steps of 50ms, providing discrete estimates of the power of neural activity at each of the frequencies of interest. Each discrete sample was mapped to the mid-point in the window at that position in the epoch. The SSVEP amplitude was then normalised by subtracting the amplitude of activity in the neighbouring frequency bins to isolate the stimulus-driven signal. The frequency bins used to extract the two SSVEP frequencies were excluded from the bins used to measure MB activity.

As mentioned in the *Procedure* section, all subjects completed calibration procedures specifically designed to identify differences in individual perception of the grating contrast caused by the different flicker frequencies. Despite these efforts to incorporate compensatory contrast adjustments based on subject-by-subject estimates of this perceptual bias, the procedures failed to identify a stable frequency biases. Of the few subjects that did appear to initially exhibit such a bias, in all but one case, it was not observed when they were asked to complete the same calibration procedures at the beginning of their next testing session. This was interpreted as evidence that exposure to the stimuli and practice on the task had reduced any bias that was initially present. Based on this assumption, once a subject had failed to show any frequency bias, no compensatory adjustment was made to the stimuli in any subsequent session, and they were not screened for a frequency bias in any subsequent session. Examination of the behavioural data after the completion of testing revealed that this had been an error as overall subjects were 1.7 times more likely to choose the grating flickering at 20 Hz, although there was no difference in the number of early responses or misses depending on the flicker frequency of the target. However, the paradigm was carefully constructed to balance the number of trials where the target was flickering at 20/25 Hz within every condition. Therefore, this issue could not have influenced any of the effects reported here. Due to differences in the amplitude of the neighbouring frequency bins used to normalise the SSVEP signal, the amplitude of the 25 Hz SSVEP was consistently greater than that of the 20 Hz signal. For this reason, the frequency of the target stimulus is included as a factor in the statistical models, but, as there were no hypotheses based on SSVEP frequency, it was not included in any interactions with variables of interest.

## Data analysis

All statistical analyses were conducted using IBM SPSS Statistics. The analyses primarily relied on mixed effects modelling, which was chosen to exploit the number of trials collected for each participant. The procedure for assessing fixed effects in a mixed effects analysis is not standardised and there is some debate about the best approach in different types of datasets and experimental designs (*Baayen et al., 2008*). However, it has been argued that for experiments using small-N samples, like the present study, the optimal approach is the estimation of degrees of freedom using a Satterthwaite approximation to compute an F-statistic (*Kuznetsova et al., 2017*; *Luke, 2017*). This method was adopted for all mixed effects analyses. A binary logistic mixed effects model was used to analyse the choice-accuracy data, all other analyses used a linear mixed effects model. A random intercept was included in all mixed effects analyses to account for the repeated-measures design. No random effects were included. Significant main effects were investigated with uncorrected pairwise comparisons.

As described above, the task included brief pulses of evidence. However, these pulses were not relevant for the hypothesis addressed in this paper. To control for the influence of the pulse, pulse type was included as a fixed effect in all behavioural and evidence-locked SSVEP mixed effects analyses. As mentioned in the previous section, the frequency of the target stimulus was also included in all mixed effects analyses of the SSVEP as a control variable. These two factors were included solely for the purposes of controlling for their effects, but without any relevance to the hypotheses being investigated, so the significance of these predictors is not reported for each analysis. As there were two lengths of testing session and participants completed different numbers of sessions, we analysed the effect of task exposure by pooling trials within-subjects and dividing them into five 'trial bins'. The first bin represents the participants' earliest exposure to the task and the final bin represents trials at the end of their participation, when they had had substantial task exposure. All trials with valid responses and reaction times greater than 100ms were included in the analyses of behavioural data and the SSVEP. As the MB and alpha analyses investigated the pre-evidence, cue-locked epoch, this reaction time restriction was not imposed, and all trials were included.

## Acknowledgements

KW was supported by an Irish Research Council Government of Ireland Postgraduate Scholarship (GOIPG/2017/1093). ROC was supported by the European Research Council Consolidator Grant IndDecision – 865474. SPK was supported by research grants from Science Foundation Ireland (15/CDA/3591) and The Wellcome Trust (219572/Z/19/Z).

## Additional information

### Competing interests

Redmond G O'Connell: Reviewing editor, *eLife*. The other authors declare that no competing interests exist.

### Funding

| Funder | Grant reference number | Author |
| --- | --- | --- |
| Irish Research Council | GOIPG/2017/1093 | Kevin Walsh |
| European Research Council | IndDecision - 865474 | Redmond G O'Connell |
| Science Foundation Ireland | 15/CDA/3591 | Simon P Kelly |
| Wellcome Trust | https://doi.org/10.35802/219572 | Simon P Kelly |

The funders had no role in study design, data collection and interpretation, or the decision to submit the work for publication. For the purpose of Open Access, the authors have applied a CC BY public copyright license to any Author Accepted Manuscript version arising from this submission.

### Author contributions

Kevin Walsh, Conceptualization, Data curation, Formal analysis, Funding acquisition, Investigation, Visualization, Methodology, Writing - original draft, Project administration, Writing – review and editing; David P McGovern, Conceptualization, Supervision, Methodology, Writing – review and editing; Jessica Dully, Conceptualization, Investigation, Writing – review and editing; Simon P Kelly, Conceptualization, Supervision, Validation, Methodology, Writing – review and editing; Redmond G O'Connell, Conceptualization, Resources, Software, Supervision, Funding acquisition, Validation, Investigation, Visualization, Methodology, Project administration, Writing – review and editing

### Author ORCIDs

Kevin Walsh http://orcid.org/0000-0002-3745-2073
David P McGovern http://orcid.org/0000-0002-5748-2827
Jessica Dully http://orcid.org/0009-0007-2556-9403

Simon P Kelly [ID] http://orcid.org/0000-0001-9983-3595
Redmond G O'Connell [ID] http://orcid.org/0000-0001-6949-2793

### Ethics

All procedures were approved by the Trinity College Dublin School of Psychology Ethics Committee (ref SPREC042020-15) and were in accordance with the Declaration of Helsinki. All participants provided informed written consent prior to testing.

Reviewer #1 (Public Review): https://doi.org/10.7554/eLife.91135.3.sa1
Reviewer #2 (Public Review): https://doi.org/10.7554/eLife.91135.3.sa2
Reviewer #3 (Public Review): https://doi.org/10.7554/eLife.91135.3.sa3
Author Response https://doi.org/10.7554/eLife.91135.3.sa4

## Additional files

### Supplementary files

• MDAR checklist

### Data availability

The datasets required to replicate the results and reproduce the figures in the manuscript are freely available, along with the task code, in the project's Open Science Framework repository (*Walsh et al., 2023*).

The following dataset was generated:

| Author(s) | Year | Dataset title | Dataset URL | Database and Identifier |
| --- | --- | --- | --- | --- |
| Walsh K, McGovern D, Dully J, Kelly S, O'Connell R | 2024 | Prior probability cues bias sensory encoding with increasing task exposure | https://osf.io/b92wm/ | Open Science Framework, 10.17605/OSF.IO/B92WM |

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
