## [Editor Report · eLife assessment]

This **important** paper sheds light on the role of expectations in perceptual decision-making. Sophisticated analyses of human EEG data provide **convincing** evidence that both motor preparation and sensory processing were affected by expectations, albeit with different time courses. These findings will be of interest to scientists interested in perception and decision-making.

---

## [Referee Report · Reviewer #1 (Public Review)]

Summary:

Walsh and colleagues investigated how cued probabilistic expectations about future stimuli may influence different stages of decision-making as implemented in the human brain. In their study, participants were provided with cues that could correctly (or incorrectly) cue which visual stimulus would be presented. These cues also predicted the motor action that would likely produce a correct judgment for that trial. In addition a 'neutral' cue was included that did not predict any particular stimulus. They report that measures of steady-state visual evoked potentials (SSVEPs, proposed to index the magnitude of visual neural activity in favour of the correct response) were smaller when the cue incorrectly predicted the upcoming image, compared to when an accurate cue or a neutral cue was presented. Their primary finding adds to an ongoing debate in the field of decision-making research about how cued expectations may influence how we make decisions.

Strengths:

This study uses a carefully-constructed experiment design and decision-making task that allows separation of multiple electroencephalographic (EEG) signals thought to track different stages of decision-making. For example, the steady-state visual evoked potential measures can be cleanly dissociated from more anterior beta-band activity over motor cortex. They also allow evaluation of how cued expectancy effects may unfold over a number of testing sessions. This is important because the most consistent evidence of expectation-related modulations of electrophysiological measures (using EEG, local field potentials or single neuron firing rates) is from studies of non-human primates that involved many days of cue-stimulus contingency learning, and there is a lack of similar work using several testing sessions in humans. Although there were several experimental conditions included in the study, careful trial-balancing was conducted to minimise biases due to incidental differences in the numbers of trials included for analyses across each condition. Performance for each individual was also carefully calibrated to maximise the possibility of identifying subtle changes in task performance by expectation and avoid floor or ceiling effects.

Weaknesses:

Although the experiment and analysis methods are cohesive and well-designed, there are some shortcomings that limit the inferences that can be drawn from the presented findings.

The first relates to the measures of SSVEPs and their relevance for decision-making in the task. In order to eliminate the influence of sporadic pulses of contrast changes that occurred during stimulus presentation, a time window of 680-975 ms post stimulus onset was used to measure the SSVEPs. As shown in the response time quantile plot in Supplementary Figure S1, a substantial portion of response times are earlier than all, or a portion of, the time period included in the SSVEP measurement window. It has also been estimated to require 70-100 ms to execute a motor action (e.g., a keypress response) following the commitment to a decision. This raises some concerns about the proportion of trials in which the contrast-dependent visual responses (indexed by the stimulus-locked SSVEPs) indexed visual input that was actually used to make the decision in a given trial. While response-locked SSVEP plots are provided, statistical analyses testing for differences during the pre-response period were not performed. Standard errors in Figure 4D (depicting differences in SSVEPs for validly and invalidly cued trials) partly overlap with zero during the pre-response time window. There is no strong evidence for clear SSVEP modulations in any specific time windows leading to the response.

In addition, an argument is made for changes in the evidence accumulation rate (called the drift rate) by stimulus expectancy, corresponding to the observed changes in SSVEP measures and differences in the sensory encoding of the stimulus. As the authors acknowledge, this inference is limited by the fact that evidence accumulation models (such as the Diffusion Decision Model) were not used to test for drift rate changes as could be determined from the behavioural data (by modelling response time distributions). Plots of response quantiles in Supplementary Figure S1 also do not show a typical pattern that indicates changes in the drift rate (i.e., larger differences between validly and invalidly cued trials for relatively slower response time quantiles). There appear to be ample numbers of trials per participant to test for drift rate changes in addition to the starting point bias captured in earlier models. Due to the very high number of trials, models could potentially be evaluated for each single participant, although modelling would be substantively complicated by effects of the pulses of contrast changes, as noted by the authors. This could be done in future work (in experiments without contrast pulses) and would provide more direct evidence for drift rate changes than the findings based on the SSVEPs, particularly due to the issues with the measurement window relating to the response times as mentioned above.

In addition, there is some uncertainty regarding how to interpret the SSVEP effects in relation to phenomena such as expectation suppression enabled via sharpening or dampening effects. The measure used in this study is marginal SSVEPs, indexing the difference in SSVEP amplitudes between relatively higher- and lower-contrast gratings (termed target and non-target gratings). The observed increase in marginal SSVEPs for validly as compared to invalidly cued trials could arise due to an increase in SSVEP amplitudes for target grating orientations, a decrease for non-target orientations, a combination of these two, or even an increase or decrease for both target and non-target SSVEPs (with a larger increase/decrease for the target or non-target orientation). Some analyses were performed to investigate predictive cueing effects on target as compared to non-target SSVEPs, but these did not provide clear evidence that favoured a specific interpretation. This should be considered when interpreting the SSVEP effects in relation to different variants of expectation suppression that have been proposed in the literature.

---

## [Referee Report · Reviewer #2 (Public Review)]

Summary:

We often have prior expectations about how the sensory world will change, but it remains an open question as to how these expectations are integrated into perceptual decisions. In particular, scientists have debated whether prior knowledge principally changes the decisions we make about the perceptual world, or directly alters our perceptual encoding of incoming sensory evidence.

The authors aimed to shed light on this conundrum by using a novel psychophysical task while measuring EEG signals that have previously been linked to either the sensory encoding or response selection phase of perceptual choice. The results convincingly demonstrate that both features of perceptual decision making are modulated by prior expectations - but that these biases in neural process emerge over different time courses (i.e., decisional signals are shaped early in learning, but biases in sensory processing are slower to emerge).

Another interesting observation unearthed in the study - though not strictly linked to this perceptual/decisional puzzle - is that neural signatures of focused attention are exaggerated on trials where participants are given neutral (i.e. uninformative) cues. This is consistent with the idea that observers are more attentive to incoming sensory evidence when they cannot rely on their expectations.

In general, I think the study makes a strong contribution to the literature, and does an excellent job of separating 'perceiving' from 'responding'. More perhaps could have been done though to separate 'perceiving' and 'responding' from 'deciding' (see below).

Strengths:

The work is executed expertly and focuses cleverly on two features of the EEG signals that can be closely connected to specific loci of the perceptual decision making process - the SSVEP which connects closely to sensory (visual) encoding, and Mu-Beta lateralisation which connects closely to movement preparation. This is a very appropriate design choice given the authors' research question.

Another advantage of the design is the use of an unusually long training regime (i.e., for humans) - which makes it possible to probe the emergence of different expectation biases in the brain over different timecourses, and in a way that may be more comparable to work with nonhuman animals (who are routinely trained for much longer than humans).

Weaknesses:

In my view, the principal shortcoming of this study is that the experimental task confounds expectations about stimulus identity with expectations about to-be-performed responses. That is, cues in the task don't just tell participants what they will (probably) see, but what they (probably) should do.

In many respects, this feature of the paradigm might seem inevitable, as if specific stimuli are not connected to specific responses, it is not possible to observe motor preparation of this kind (e.g., de Lange, Rahnev, Donner & Lau, 2013 - JoN).

However, the theoretical models that the authors focus on (e.g., drift diffusion models) are models of decision (i.e., commitment to a proposition about the world) as much as they are models of choice (i.e., commitment to action). Expectation researchers interested in these models are often interested in asking whether predictions influence perceptual processing, perceptual decision and/or response selection stages (e.g., Feuerriegel, Blom & Hoogendorn, 2021 - Cortex), and other researchers have shown that parameters like drift bias and start point bias can be shifted in paradigms where observers cannot possibly prepare a response (e.g., Thomas, Yon, de Lange & Press, 2020 - Psych Sci).

The present paradigm used by Walsh et al makes it possible to disentangle sensory processing from later decisional processes, but it blurs together the processes of deciding about the stimulus and choosing/initiating the response. This ultimately limits the insights we can draw from this study - as it remains unclear whether rapid changes in motor preparation we see reflect rapid acquisition of new decision criterion or simple cue-action learning. I think this would be important for comprehensively testing the models the authors target - and a good avenue for future work.

In revising the manuscript after an initial round of revisions, the authors have done a good job of acknowledging these complexities - and I don't think that any of these outstanding scientific puzzles detract from the value of the paper as a whole.

---

## [Referee Report · Reviewer #3 (Public Review)]

Observers make judgements about expected stimuli faster and more accurately. How expectations facilitate such perceptual decisions remains an ongoing area of investigation, however, as expectations may exert their effects in multiple ways. Expectations may directly influence the encoding of sensory signals. Alternatively (or additionally), expectations may influence later stages of decision-making, such as motor preparation, when they bear on the appropriate behavioral response.

In the present study, Walsh and colleagues directly measured the effect of expectations on sensory and motor signals by making clever use of the encephalogram (EEG) recorded from human observers performing a contrast discrimination task. On each trial, a predictive cue indicated which of two superimposed stimuli would likely be higher contrast and, therefore, whether a left or right button press was likely to yield a correct response. Deft design choices allowed the authors to extract both contrast-dependent sensory signals and motor preparation signals from the EEG. The authors provide compelling evidence that, when predictive cues provide information about both a forthcoming stimulus and the appropriate behavioral response, expectation effects are immediately manifest in motor preparation signals and only emerge in sensory signals after extensive training.

Future work should attempt to reconcile these results with related investigations in the field. As the authors note, several groups have reported expectation-induced modulation of sensory signals (using both fMRI and EEG/MEG) on shorter timescales (e.g. just one or two sessions of a few hundred trials, versus the intensive multi-session study reported here). One interesting possibility is that perceptual expectations are not automatic but demand the deployment of feature-based attention, while motor preparation is comparatively less effortful and so dominates when both sources of information are available, as in the present study. This hypothesis is consistent with the authors' thoughtful analysis showing decreased neural signatures of attention over posterior electrodes following predictive cues. Therefore, observing the timescale of sensory effects using the same design and methods (facilitating direct comparison with the present work), but altering task demands slightly such that cues are no longer predictive of the appropriate behavioral response, could be illuminating.

---

## [Author Response]

The following is the authors’ response to the original reviews.

**Reviewer #1**
Strengths:This study uses a carefully constructed experiment design and decision-making task that allows separation of multiple electroencephalographic (EEG) signals thought to track different stages of decision-making. For example, the steady-state visual evoked potential measures can be cleanly dissociated from more anterior beta-band activity over the motor cortex. They also allow evaluation of how cued expectancy effects may unfold over a number of testing sessions. This is important because the most consistent evidence of expectation-related modulations of electrophysiological measures (using EEG, local field potentials, or single neuron firing rates) is from studies of nonhuman primates that involved many days of cue-stimulus contingency learning, and there is a lack of similar work using several testing sessions in humans. Although there were several experimental conditions included in the study, careful trial-balancing was conducted to minimise biases due to incidental differences in the number of trials included for analyses across each condition. Performance for each individual was also carefully calibrated to maximise the possibility of identifying subtle changes in task performance by expectation and avoid floor or ceiling effects.

We would like to thank Reviewer 1 for these very positive comments.

Weaknesses:Although the experiment and analysis methods are cohesive and well-designed, there are some shortcomings that limit the inferences that can be drawn from the presented findings.Comment #1The first relates to the measures of SSVEPs and their relevance for decision-making in the task. In order to eliminate the influence of sporadic pulses of contrast changes that occurred during stimulus presentation, a time window of 680-975 ms post-stimulus onset was used to measure the SSVEPs. The mean response times for the valid and neutral cues were around 850-900 ms for correct responses, and within the same time window for errors in the invalid cue condition. In addition, a large portion of response times in perceptual decision-making tasks are substantially faster than the mean due to right-skewed response time distributions that are typically observed. As it has also been estimated to require 70-100 ms to execute a motor action (e.g., a keypress response) following the commitment to a decision. This raises some concerns about the proportion of trials in which the contrast-dependent visual responses (indexed by the SSVEPs) indexed visual input that was actually used to make the decision in a given trial. Additional analyses of SSVEPs that take the trial-varying pulses into account could be run to determine whether expectations influenced visual responses earlier in the trial.

The reviewer raises a very valid point and, indeed, it is an issue that we grappled with in our analyses. Actually, in this study, the RT distributions were not right-skewed, but appear to be relatively normal (RT distributions shown below). This is something that we have previously observed when using tasks that involve an initial zero-evidence lead in at the start of each trial which means that participants cannot start accumulating at stimulus onset and must rely on their knowledge of the lead-in duration to determine when the physical evidence has become available (e.g. Kelly et al 2021, Nat Hum Beh). We agree that it is important to establish whether the reported SSVEP modulations occur before or after choice commitment. In our original submission we had sought to address this question through our analysis of the response-locked ‘difference SSVEP’. Figure 4D clearly indicates that the cue modulations are evident before as well as after response.

However, we have decided to include an additional Bayesian analysis of the response-locked signal to offer more evidence that the cue effect is not a post-response phenomenon.

Manuscript Changes

To quantify the evidence that the cue effect was not driven by changes in the signal after the response, we ran Bayesian one-way ANOVAs on the SSVEP comparing the difference across cue conditions before and after the response. If the cue effect only emerged after the response, we would expect the difference between invalid and neutral or invalid and valid cues to increase in the post-response window. There was no compelling evidence of an increase in the effect when comparing invalid to neutral (BF10 = 1.58) or valid cues (BF10 = 0.32).

Comment #2Presenting response time quantile plots may also help to determine the proportions of motor responses (used to report a decision) that occurred during or after the SSVEP measurement window.

We agree that it may be helpful for the reader to be able to determine the proportion of responses occurring at different phases of the trial, so we have included the requested response time quantile plot (shown below) as a supplementary figure.

**Author response image 1. sa4fig1:** Reaction time quantiles across cue conditions. The plot illustrates the proportion of trials where responses occurred at different stages of the trial. The SSVEP analysis window is highlighted in purple.

Comment #3In addition, an argument is made for changes in the evidence accumulation rate (called the drift rate) by stimulus expectancy, corresponding to the observed changes in SSVEP measures and differences in the sensory encoding of the stimulus. This inference is limited by the fact that evidence accumulation models (such as the Diffusion Decision Model) were not used to test for drift rate changes as could be determined from the behavioural data (by modelling response time distributions). There appear to be ample numbers of trials per participant to test for drift rate changes in addition to the starting point bias captured in earlier models. Due to the very high number of trials, models could potentially be evaluated for each single participant. This would provide more direct evidence for drift rate changes than the findings based on the SSVEPs, particularly due to the issues with the measurement window relating to the response times as mentioned above.

The focus of the present study was on testing for sensory-level modulations by predictive cues, rather than testing any particular models. Given that the SSVEP bears all the characteristics of a sensory evidence encoding signal, we believe it is reasonable to point out that its modulation by the cues would very likely translate to a drift rate effect. But we do agree with the reviewer that any connection between our results and previously reported drift rate effects can only be confirmed with modelling and we have tried to make this clear in the revised text. We plan to comprehensively model the data from this study in a future project. While we do indeed have the benefit of plenty of trials, the modelling process will not be straightforward as it will require taking account of the pulse effects which could have potentially complicated, non-linear effects. In the meantime, we have made changes to the text to qualify the suggestion and stress that modelling would be necessary to determine if our hypothesis about a drift rate effect is correct.

Manuscript Changes

(Discussion): [...] We suggest that participants may have been able to stabilise their performance across task exposure, despite reductions in the available sensory evidence, by incorporating the small sensory modulation we detected in the SSVEP. This would suggest that the decision process may not operate precisely as the models used in theoretical work describe. Instead, our study tentatively supports a small number of modelling investigations that have challenged the solitary role of starting point bias, implicating a drift bias (i.e. a modulation of the evidence before or upon entry to the decision variable) as an additional source of prior probability effects in perceptual decisions (Dunovan et al., 2014; Hanks et al., 2011; Kelly et al., 2021; van Ravenzwaaij et al., 2012 Wyart et al., 2012) and indicates that these drift biases could, at least partly, originate at the sensory level. However, this link could only be firmly established with modelling in a future study.

**Recommendations For The Authors:**
Comment #4The text for the axis labels and legends in the figures is quite small relative to the sizes of the accompanying plots. I would recommend to substantially increase the sizes of the text to aid readability.

Thank you for this suggestion. We have increased the size of the axis labels and made the text in the figure legends just 1pt smaller than the text in the main body of the manuscript.

Comment #5

It is unclear if the scalp maps for Figure 5 (showing the mu/beta distributions) are on the same scale or different scales. I assume they are on different scales (adjusted to the minimum/maximum within each colour map range), as a lack of consistent signals (in the neutral condition) would be expected to lead to a patchy pattern on the scalp as displayed in that figure (due to the colour range shrinking to the degree of noise across electrodes). I would recommend to include some sort of colour scale to show that, for example, in the neutral condition there are no large-amplitude mu/ beta fluctuations distributed somewhat randomly across the scalp.

Thank you to the reviewer for pointing this out. They were correct, the original topographies were plotted according to their own scale. The topographies in Figure 5 have now been updated to put them on a common scale and we have included a colour bar (as shown below). The caption for Figure 5 has also been updated to confirm that the topos are on a common scale.

**Author response image 2. sa4fig2:** 

Manuscript Changes

(Figure 5 Caption): [...] The topography of MB activity in the window - 200:0 ms before evidence onset is plotted on a common scale for neutral and cued conditions separately.

Comment #6In Figure 2, the legend is split across the two panels, despite the valid/invalid/neutral legend also applying to the first panel. This gives an initial impression that the legend is incomplete for the first panel, which may confuse readers. I would suggest putting all of the legend entries in the first panel, so that all of this information is available to readers at once.

We are grateful to the reviewer for spotting this. Figure 2 has been updated so that the full legend is presented in the first panel, as shown below.

**Author response image 3. sa4fig3:** 

Comment #7Although linear mixed-effects models (using Gaussian families) for response times are standard in the literature, they incorrectly specify the distributions of response times to be Gaussian instead of substantially right-skewed. Generalised linear mixed-effects models using gamma families and identity functions have been shown to more accurately model distributions of response times (see Lo and Andrews, 2015. Frontiers in Psychology). The authors may consider using these models in line with good practice, although it might not make a substantial difference relating to the patterns of response time differences.

We appreciate this thoughtful comment from Reviewer 1. Although RT distributions are often right skewed, we have previously observed that RT distributions can be closer to normal when the trial incorporates a lead-in phase with no evidence (e.g. Kelly et al 2021, Nat Hum Beh). Indeed, the distributions we observed in this study were markedly Gaussian (as shown in the plot below). Given the shape of these distributions and the reviewer’s suggestion that adopting alternative models may not lead to substantial differences to our results, we have decided to leave the mixed effects models as they are in the manuscript, but we will take note of this advice in future work.

**Author response image 4. sa4fig4:** 

**Reviewer #2**
Strengths:The work is executed expertly and focuses cleverly on two features of the EEG signals that can be closely connected to specific loci of the perceptual decision-making process - the SSVEP which connects closely to sensory (visual) encoding, and Mu-Beta lateralisation which connects closely to movement preparation. This is a very appropriate design choice given the authors' research question.Another advantage of the design is the use of an unusually long training regime (i.e., for humans) - which makes it possible to probe the emergence of different expectation biases in the brain over different timecourses, and in a way that may be more comparable to work with nonhuman animals (who are routinely trained for much longer than humans).

We are very grateful for these positive comments from Reviewer 2.

Weaknesses:In my view, the principal shortcoming of this study is that the experimental task confounds expectations about stimulus identity with expectations about to-be-performed responses. That is, cues in the task don't just tell participants what they will (probably) see, but what they (probably) should do.In many respects, this feature of the paradigm might seem inevitable, as if specific stimuli are not connected to specific responses, it is not possible to observe motor preparation of this kind (e.g., de Lange, Rahnev, Donner & Lau, 2013 - JoN).However, the theoretical models that the authors focus on (e.g., drift-diffusion models) are models of decision (i.e., commitment to a proposition about the world) as much as they are models of choice (i.e., commitment to action). Expectation researchers interested in these models are often interested in asking whether predictions influence perceptual processing, perceptual decision, and/ or response selection stages (e.g., Feuerriegel, Blom & Hoogendorn, 2021 - Cortex), and other researchers have shown that parameters like drift bias and start point bias can be shifted in paradigms where observers cannot possibly prepare a response (e.g., Thomas, Yon, de Lange & Press, 2020 - Psych Sci).The present paradigm used by Walsh et al makes it possible to disentangle sensory processing from later decisional processes, but it blurs together the processes of deciding about the stimulus and choosing/initiating the response. This ultimately limits the insights we can draw from this study - as it remains unclear whether rapid changes in motor preparation we see reflect rapid acquisition of new decision criterion or simple cue-action learning. I think this would be important for comprehensively testing the models the authors target - and a good avenue for future work.

Thank you to Reviewer 2 for these observations. We adopted this paradigm because it is typical of the perceptual decision making literature and our central focus in this study was to test for a sensory-level modulation as a source of a decision bias. We are pleased that the Reviewer agrees that the paradigm successfully disentangles sensory encoding from later decisional processes since this was our priority. However, we agree with Reviewer 2 that because the response mapping was known to the participants, the cues predicted both the outcome of the perceptual decision (“Is this a left- or right-tilted grating?”) and the motor response that the participant should anticipate making (“It’s probably going to be a left click on this trial”). They are correct that this makes it difficult to know whether the changes in motor preparation elicited by the predictive cues reflect action-specific preparation or a more general shift in the boundaries associated with the alternate perceptual interpretations. We fully agree that it remains an interesting and important question and in our future work we hope to conduct investigations that better dissect the distinct components of the decision process during prior-informed decisions. In the interim, we have made some changes to the manuscript to reflect the Reviewer’s concerns and better address this limitation of the study design (these are detailed in the response to the comment below).

**Recommendations For The Authors:**
Comment #8As in my public review, my main recommendation to the authors is to think a bit more in the presentation of the Introduction and Discussion about the difference between 'perceiving', 'deciding', and 'responding'.The paper is presently framed in terms of the debates around whether expectations bias decision or bias perception - and these debates are in turn mapped onto different aspects of the driftdiffusion model. Biases in sensory gain, for instance, are connected to biases in the drift rate parameter, while decisional shifts are connected to parameters like start points.In line with this kind of typology, the authors map their particular EEG signals (SSVEP and MB lateralisation) onto perception and decision. I see the logic, but I think the reality of these models is more nuanced.In particular, strictly speaking, the process of evidence accumulation to bound is the formation of a 'decision' (i.e., a commitment to having seen a particular stimulus). Indeed, the dynamics of this process have been beautifully described by other authors on this paper in the past. Since observers in this task simultaneously form decisions and prepare actions (because stimuli and responses are confounded) it is unclear whether changes in motor preparation are reflecting changes in what perceivers 'decide' (i.e., changes in what crosses the decision threshold) or what they 'do' (i.e., changes in the motor response threshold). This is particularly important for the debate around whether expectations change 'perception' or 'decision' because - in some accounts - is the accumulation of evidence to the bound that is hypothesised to cause the perceptual experience observers actually have (Pereira, Perrin & Faivre, 2022 - TiCS). The relevant 'bound' here though is not the bound to push the button, but the bound for the brain to decide what one is actually 'seeing'.I completely understand the logic behind the authors' choices, but I would have liked more discussion of this issue. In particular, it seems strange to me to talk about the confounding of stimuli and responses as a particular 'strength' of this design in the manuscript - when really it is a 'necessary evil' for getting the motor preparation components to work. Here is one example from the Introduction:"While some have reported expectation effects in humans using EEG/MEG, these studies either measured sensory signals whose relevance to the decision process is uncertain (e.g. Blom et al., 2020; Solomon et al., 2021; Tang et al., 2018) and/or used cues that were implicit or predicted a forthcoming stimulus but not the correct choice alternative (e.g. Aitken et al., 2020; Feuerriegel et al., 2021b; Kok et al., 2017). To assess whether prior probabilities modulate sensory-level signals directly related to participants' perceptual decisions, we implemented a contrast discrimination task in which the cues explicitly predicted the correct choice and where sensory signals that selectively trace the evidence feeding the decision process could be measured during the process of deliberation."I would contend that this design allows you to pinpoint signals related to participant's 'choices' or 'actions' but not necessarily their 'decisions' in the sense outlined above.As I say though, I don't think this is fatal and I think the paper is extremely interesting in any case. But I think it would be strengthened if some of these nuances were discussed a bit more explicitly, as a 'perceptual decision' is more than pushing a button. Indeed, the authors might want to consider discussing work that shows the neural overlap between deciding and acting breaks down when Ps cannot anticipate which actions to use to report their choices ahead of time (Filimon, Philiastides, Nelson, Kloosterman & Heekeren, 2013 - JoN) and/or work which has combined expectations with drift diffusion modelling to show how expectations change drift bias (Yon, Zainzinger, de Lange, Eimer & Press, 2020 - JEP:General) and/or start bias (Thomas, Yon, de Lange & Press, 2020 - Psych Sci) even when Ps cannot prepare a motor response ahead of time.

While our focus was on testing for sensory-level modulations, we think the question of whether the motor-level effects we observed are attributable to the task design or represents a more general perceptual bound adjustment is an important question for future research. In our previous work, we have examined this distinction between abstract, movement-independent evidence accumulation (indexed by the centro-parietal positivity, CPP) and response preparation in detail. The CPP has been shown to trace evidence accumulation irrespective of whether the sensory alternatives are associated with a specific response or not (Twomey et al 2016, J Neurosci). When speed pressure is manipulated in tasks with fixed stimulus-response mappings we have found that the CPP undergoes systematic adjustments in its pre-response amplitude that closely accord with the starting-level modulations observed in mu/beta, suggesting that motor-level adjustments do still translate to differences at the perceptual level under these task conditions (e.g. Kelly et al 2021, Nat Hum Beh; Steinemann et al., 2018, Nat Comms). We have also observed that the CPP and mu-beta exhibit corresponding adjustments in response to predictive cues (Kelly et al., 2021) that are consistent with both a starting-point shift and drift rate bias. However, the Kelly et al. study did not include a signature of sensory encoding and therefore could not test for sensory-level modulations.

We have added some remarks to the discussion to acknowledge this issue with the interpretation of the preparatory shifts in mu-beta activity we observed when the predictive cues were presented, and we have included references to the papers that the reviewer helpfully provided. We have also offered some additional consideration of the features of the task design that may have influenced the SSVEP results.

Manuscript Changes

An implication of using cues that predict not just the upcoming stimulus, but the most likely response, is that it becomes difficult to determine if preparatory shifts in mu-beta (MB) activity that we observed reflect adjustments directly influencing the perceptual interpretation of the stimulus or simply preparation of the more probable action. When perceptual decisions are explicitly tied to particular modes of response, the decision state can be read from activity in motor regions associated with the preparation of that kind of action (e.g. de Lafuente et al., 2015; Ding & Gold, 2012; Shadlen & Newsome, 2001; Romo et al., 2004), but these modules appear to be part of a constellation of decision-related areas that are flexibly recruited based on the response modality (e.g. Filimon et al., 2013). When the response mapping is withheld or no response is required, MB no longer traces decision formation (Twomey et al., 2015), but an abstract decision process is still readily detectable (e.g. O’Connell et al., 2012), and modelling work suggests that drift biases and starting point biases (Thomas et al., 2020; Yon et al., 2021) continue to influence prior-informed decision making. While the design of the present study does not allow us to offer further insight about whether the MB effects we observed were inherited from strategic adjustments at this abstract level of the decision process, we hope to conduct investigations in the future that better dissect the distinct components of prior-informed decisions to address this question.

Several other issues remain unaddressed by the present study. One, is that it is not clear to what extent the sensory effects may be influenced by features of the task design (e.g. speeded responses under a strict deadline) and if these sensory effects would generalise to many kinds of perceptual decision-making tasks or whether they are particular to contrast discrimination.

Comment #9On a smaller, unrelated point - I thought the discussion in the Discussion section about expectation suppression was interesting, but I did not think it was completely logically sound. The authors suggest that they may see relative suppression (rather than enhancement) of their marginal SSVEP under a 'sharpening' account because these accounts suggest that there is a relative suppression of off-channel sensory units, and there are more off-channel sensory units than onchannel sensory units (i.e., there are usually more possibilities we don't expect than possibilities that we do, and suppressing the things we don't expect should therefore yield overall suppression).However, this strikes me as a non-sequitur given that the marginal SSVEP only reflects featurespecific visual activity (i.e., activity tuned to one of the two grating stimuli used). The idea that there are more off-channel than on-channel units makes sense for explaining why we would see overall signal drops on expected trials e.g., in an entire visual ROI in an fMRI experiment. But surely this explanation cannot hold in this case, as there is presumably an equal number of units tuned to each particular grating?My sense is that this possibility should probably be removed from the manuscript - and I suspect it is more likely that the absence of a difference in marginal SSVEP for Valid vs Neutral trials has more to do with the fact that participants appear to be especially attentive on Neutral trials (and so any relative enhancement of feature-specific activity for expected events is hard to detect against a baseline of generally high-precision sensory evidence on these highly attentive, neutral trials).

We thank the reviewer for flagging that we did not clearly articulate our thoughts in this section of the manuscript. Our primary purpose in mentioning this sharpening account was simply to point out that, where at first blush our results seem to conflict with expectation suppression effects in the fMRI literature, the sharpening account provides an explanation that can reconcile them. In the case of BOLD data, the sharpening account proposes that on-channel sensory units are boosted and off-channel units are suppressed and, due to the latter being more prevalent, this leads to an overall suppression of the global signal. In the case of the SSVEP, the signal isolates just the onunits and so the sharpening account would predict that when there is a valid cue, the SSVEP signal associated with the high-contrast, expected stimulus should be boosted and the SSVEP signal associated with the low-contrast, unexpected stimulus should be weakened; this would result in a larger difference between these signals and therefore, a larger ‘marginal SSVEP’. Conversely, when there is an invalid cue, the SSVEP signal associated with the, now unexpected, high-contrast stimulus should be relatively weakened and the SSVEP signal associated with the expected, but low-contrast stimulus should be relatively boosted; this would result in a smaller difference between these signals and therefore, a lower amplitude marginal SSVEP. We do not think that this account needs to make reference to any channels beyond those feature-specific channels driving the two SSVEP signals. Again our central point is simply that the sharpening account offers a means of reconciling our SSVEP findings with expectation suppression effects previously reported in the fMRI literature.

We suspect that this was not adequately explained in the discussion. We have adjusted the way this section is phrased to make it clear that we are not invoking off-channel activity to explain the SSVEP effect we observed and we thank the Reviewer for pointing out that this was unclear in the original text.

Manuscript Changes

An alternative account for expectation suppression effects, which is consistent with our SSVEP results, is that they arise, not from a suppression of expected activity, but from a ‘sharpening’ effect whereby the response of neurons that are tuned to the expected feature are enhanced while the responses of neurons tuned to unexpected features are suppressed (de Lange et al., 2018). On this account, the expectation suppression commonly reported in fMRI studies arises because voxels contain intermingled populations with diverse stimulus preferences and the populations tuned to the unexpected features outnumber those tuned to the expected feature. In contrast to these fMRI data, the SSVEP represents the activity of sensory units driven at the same frequency as the stimulus, and thus better isolates the feature-specific populations encoding the task-relevant sensory evidence. Therefore, according to the sharpening account, an invalid cue would have enhanced the SSVEP signal associated with the low contrast grating and weakened the SSVEP signal associated with the high contrast grating. As this would result in a smaller difference between these signals, and therefore, a lower amplitude marginal SSVEP compared to the neutral cue condition, this could explain the effect we observed.

**Reviewer #3**
Observers make judgements about expected stimuli faster and more accurately. How expectations facilitate such perceptual decisions remains an ongoing area of investigation, however, as expectations may exert their effects in multiple ways. Expectations may directly influence the encoding of sensory signals. Alternatively (or additionally), expectations may influence later stages of decision-making, such as motor preparation, when they bear on the appropriate behavioral response.In the present study, Walsh and colleagues directly measured the effect of expectations on sensory and motor signals by making clever use of the encephalogram (EEG) recorded from human observers performing a contrast discrimination task. On each trial, a predictive cue indicated which of two superimposed stimuli would likely be higher contrast and, therefore, whether a left or right button press was likely to yield a correct response. Deft design choices allowed the authors to extract both contrast-dependent sensory signals and motor preparation signals from the EEG. The authors provide compelling evidence that, when predictive cues provide information about both a forthcoming stimulus and the appropriate behavioral response, expectation effects are immediately manifest in motor preparation signals and only emerge in sensory signals after extensive training.Future work should attempt to reconcile these results with related investigations in the field. As the authors note, several groups have reported expectation-induced modulation of sensory signals (using both fMRI and EEG/MEG) on shorter timescales (e.g. just one or two sessions of a few hundred trials, versus the intensive multi-session study reported here). One interesting possibility is that perceptual expectations are not automatic but demand the deployment of feature-based attention, while motor preparation is comparatively less effortful and so dominates when both sources of information are available, as in the present study. This hypothesis is consistent with the authors' thoughtful analysis showing decreased neural signatures of attention over posterior electrodes following predictive cues. Therefore, observing the timescale of sensory effects using the same design and methods (facilitating direct comparison with the present work), but altering task demands slightly such that cues are no longer predictive of the appropriate behavioral response, could be illuminating.

We would like to thank Reviewer 3 for their positive comments and thoughtful suggestions for future work.

**Recommendations For The Authors:**
Comment #10In the methods, the term 'session' is used early on but only fleshed out at the end of the 'Procedure' subsection and never entirely explained (e.g., did sessions take place over multiple days?). A brief sentence laying this out early on, perhaps in 'Participants' after the (impressive) trial counts are reported, might be helpful.

Thank you to Reviewer 3 for pointing out that this was not clear in the original draft. We have amended the text in the Methods section to better explain the relationship between sessions, days, and trial bins.

Manuscript Changes

(Methods - Participants): [...] All procedures were approved by the Trinity College Dublin School of Psychology Ethics Committee and were in accordance with the Declaration of Helsinki. Participants completed between 4 and 6 testing sessions, each on a different day. While the sample size was small, on average, participants completed 5750 (SD = 1066) trials each.

(Methods - Data Analysis): [...] As there were two lengths of testing session and participants completed different numbers of sessions, we analysed the effect of task exposure by pooling trials within-subjects and dividing them into five ‘trial bins’. The first bin represents the participants’ earliest exposure to the task and the final bin represents trials at the end of their participation, when they had had substantial task exposure. All trials with valid responses and reaction times greater than 100 ms were included in the analyses of behavioural data and the SSVEP.

Comment #11On a related note: participants completed a variable number of trials/sessions. To facilitate comparison across subjects, training effects are reported by dividing each subject's data into 5 exposure bins. This is entirely reasonable but does leave the reader wondering about whether you found any effects of rest or sleep between sessions.

We agree with the reviewer that this is an interesting question that absolutely merits further investigation. As different participants completed different numbers of sessions, different session lengths, and had variable gaps between their sessions, we do not think a per-session analysis would be informative. We think it may be better addressed in a future study, perhaps one with a larger sample where we could collect data specifically about sleep and more systematically control the intervals between testing sessions.

Comment #12Fig 2B: the 'correct' and 'neutral' labels in the legend are switched

Thank you to the reviewer for spotting that error, the labels in Figure 2 have been corrected.

Comment #13Fig 4B: it's a bit difficult to distinguish which lines are 'thick' and 'thin'

We have updated Figure 4.B to increase the difference in line thickness between the thick and thin lines (as shown below).

**Author response image 5. sa4fig5:** 

Comment #14Fig 4C: missing (I believe?) the vertical lines indicating median reaction time

We have updated Figure 4.C to include the median reaction times.

**Author response image 6. sa4fig6:**